

# Redescription of four *Epiperipatus* species with an update on the distribution of *Epiperipatus acacioi* (Marcus & Marcus, 1955)

Cristiano Sampaio Costa[1], Robson de Almeida Zampaulo[2], Santelmo Vasconcelos[3], Michele Molina[3], Igor Cizauskas[4] and Ricardo Pinto-da-Rocha[5]

[1] Colegiado de Ciências Biológicas, Universidade de Pernambuco, Petrolina, Pernambuco, Brazil
[2] Gerência de Licenciamento Ambiental e Espeleologia, Vale S.A., Nova Lima, Minas Gerais, Brazil
[3] Instituto Tecnológico Vale, Belém, Pará, Brazil
[4] Organização de Apoio à Pesquisa da Biodiversidade, São Paulo, São Paulo, Brazil
[5] Instituto de Biociências, Departamento de Zoologia, Universidade de São Paulo, São Paulo, São Paulo, Brazil

Corresponding authors
Cristiano Sampaio Costa,
cristiano.costa@upe.br,
csampaioc@gmail.com
Santelmo Vasconcelos,
santelmo.vasconcelos@itv.org

## ABSTRACT

Due to recent phylogenetic studies on Neopatida over the last ten years, the genus *Epiperipatus* has become the most diverse within Peripatidae. Such an expansion occurred due to nomenclatural acts based on evidence from anatomical characters that had not been well supported for genera and species included in the last *Epiperipatus* revision. Among these species are *Epiperipatus brasiliensis* (Bouvier, 1900), *E. acacioi* (*Marcus & Marcus, 1955*), *E. cratensis Brito et al., 2010* and *Peripatus bouvieri Fuhrmann, 1913*. Here, we provide the redescription of these three species previously included in *Epiperipatus*, besides presenting *Epiperipatus bouvieri* as a new combination. We extended the distribution of *E. acacioi* for the Serra da Moeda in the municipalities of Rio Acima, Nova Lima, and Itabirito (Minas Gerais, Brazil). The molecular data showed that specimens from these locations are closely related to *E. acacioi*, forming a clade deeply nested within the *Epiperipatus* group. Hitherto, the distribution of the species was restricted to the Estação Ecológica do Tripuí in the municipality of Ouro Preto. However, based on our results, the natural porosity of the rocks associated with the iron ore deposits of this region may have allowed the dispersal of the species along the mountains in this region for tens of kilometers.

## INTRODUCTION

The family Peripatidae *Audouin & Milne-Edwards, 1832*, with around 80 nominal species, has 92% of its current diversity distributed in the Neotropics, with most of the taxa occurring in central-northern South America, Central America, and the Caribbean, extending along the western coast in the central part of Mexico (*Brusca, Giribet & Moore, 2023*). This diversity is mainly present in tropical rainforests, with a high concentration of species in Brazil, thanks to recent efforts to describe onychophorans from the country,

where the diversity has duplicated in the last 18 years (*Oliveira & Wieloch, 2005*; *Brito et al., 2010*; *Oliveira et al., 2011*; *Oliveira et al., 2015*; *Costa, Chagas-Junior & Pinto-da-Rocha, 2018*; *Costa, Mendes & Giupponi, 2023*). However, the present state of onychophoran systematics still needs improvement to attend to the increasing diversity of the groups, as there are still cases of genera based on faulty systems without rigorous phylogenetic tests (*Costa, Giribet & Pinto-da-Rocha, 2021*).

The introduction of molecular inferences has shed light on the internal relationships in Neopatida. Recently proposed taxonomic changes have extended the diversity of *Epiperipatus Clark, 1913*, turning it into the most diverse genus within Onychophora with 33 species (*Costa, Mendes & Giupponi, 2023*; *Oliveira, 2023*) and inhabiting regions in Central and Eastern South America. Furthermore, Peripatidae has been widely studied in the last eight years, with taxonomic revisions and phylogenetic analyses expanding the family diversity based on phylogenetic analyses (*Cunha et al., 2017*; *Giribet et al., 2018*; *Oliveira, 2023*), as in the case of the revision of the Brazilian species *E. brasiliensis* (*Bouvier, 1899*), *E. acacioi* (*Marcus & Marcus, 1955*), *E. cratensis* (*Brito et al., 2010*), and *E. sucuriuensis* (*Oliveira et al., 2015*). However, there is still a significant gap between the current diversity of Peripatidae and the extensive sampling volume representing this family in museum collections, fomenting a debate on the validation of Neopatida genera.

Additionally, the knowledge of Brazilian onychophorans has improved since the beginning of the 21st century, considering that the numbers of described species considerably increased in almost two decades. Currently, 21 species are known from Brazil, with most being endemic and threatened (*Costa, Chagas-Junior & Pinto-da-Rocha, 2018*; *Costa, Medeiros & Zampaulo, 2023*; *Costa, Mendes & Giupponi, 2023*). *Epiperipatus acacioi* represents the first documented description of onychophorans from Minas Gerais, garnering significant attention due to its wide-ranging applicability in various scientific fields, besides serving as a flagship invertebrate species, supporting the implementation of the conservation area Estação Ecológica do Tripuí (*IEF–Instituto Estadual de Florestas, 1995*). However, it was only nearly fifty years later that our understanding of the onychophoran diversity in this region has been expanded. Minas Gerais encompasses the Brazilian biomes of Cerrado and Atlantic Forest (*Morrone, 2014*), including several contrasting ecosystems such as forests, mountains, and caves, presenting a great diversity of habitats for new sampling. Accordingly, Brazil's significant increase in cave fauna inventories over the last two decades has yielded new records for these invertebrates.

Regarding Brazilian onychophorans, it is worth noting that all named species are epigean, with only two troglobitic records reported in central Brazil (*Cordeiro, Borghezan & Trajano, 2014*), which are yet to be formally described. The presence of troglomorphic characteristics in Onychophora remains a conundrum, as these adaptations were seldom reported, with different species being occasionally found in entrances and interiors of caves in various Brazilian states, including Minas Gerais, Mato Grosso, Mato Grosso do Sul, Goiás, Pará and Tocantins (*Sampaio-Costa, Chagas-Junior & Baptista, 2009*; *Trajano & Bichuette, 2010*; *Cordeiro, Borghezan & Trajano, 2014*; *Costa, 2016*; *Costa, Medeiros & Zampaulo, 2023*). Close to the Estação Ecológica do Tripuí, in Rio Acima (Minas Gerais), we have encountered an intriguing case involving four specimens of *E. acacioi* (Fig. 1). A

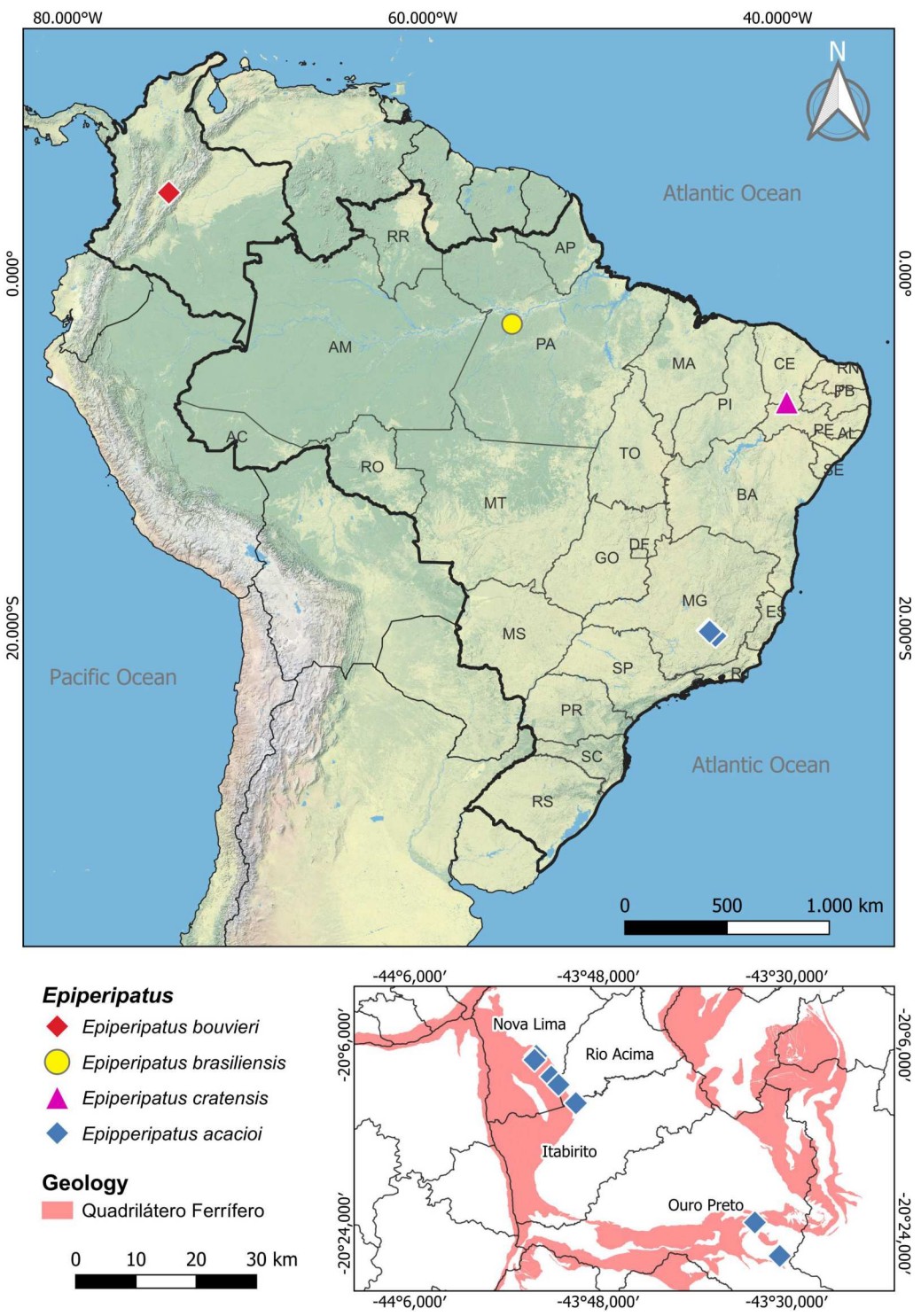

**Figure 1** **Map of distribution of the species of *Epiperipatus* studied here.** The distribution of *Epiperipatus acacioi* in the state of Minas Gerais is highlighted on the smaller map at the bottom right. Records from Nova Lima and Rio Acima belong to the new specimens of *E. acacioi*. Note the distance from the new record and the type locality.

further investigation based on morphology and molecular data has revealed the first record of *E. acacioi* outside the Estação Ecológica do Tripuí. Therefore, the present work aims to present the results of the redescription of four species of *Epiperipatus*, based on findings by *Costa (2016)* providing images documenting the main body characteristics either not mentioned in the seminal manuscripts or previously not documented under scanning electron microscopy (SEM), as well as expanding the distribution data of *E. acacioi*.

## MATERIAL AND METHODS

### Specimen acquisition

We examined a total of 67 specimens from the following institutions (and their abbreviations): Museu Nacional, Universidade Federal do Rio de Janeiro, Rio de Janeiro, Brazil (MNRJ); Museu de Zoologia da Universidade de São Paulo, São Paulo, Brazil (MZUSP); Universidade Federal de Minas Gerais, Belo Horizonte, Brazil (UFMG); Universidade Regional do Cariri (LZ-URCA); Universidad Nacional de Colombia (ICN-ONY); and Natural History Museum, London, United Kingdom (NHM). For color and body patterns while alive, we photographed the samples with a Sony Cybershot DSC-HX1 with built-in flash or a Canon EOS Rebel XS with a macro lens and ring flash circular cameras. We described the colors following the standard names of the NBS/ISCC Color System (see *Kelly, 1958*) and used them in descriptions. Synonyms were arranged in chronological order. Newly collected specimens were in accordance with the sampling permits 32882-1 and 34803-2, approved by ICMBio/MMA (Brazilian Ministry of Environment).

### Morphological redescription

Color and morphology patterns of specimens were described primarily following *Bouvier (1905)*, *Marcus & Marcus (1955)*, *Oliveira, Wieloch & Mayer (2010)*, *Brito et al. (2010)*, and *Costa (2016)*. We have focused on selected characters, as discussed by *Costa (2016)*, including the examination of patterns, and their variations in the dorsal body region, antennal rings, morphology of dorsal papillae, patterns of the jaws, number of legs and spinous pads, position of nephridial tubercles, evidence of anal glands, and patterns of the crural papillae.

For the investigations of morphological characters, the specimens were examined with a Leica S8AP0 stereomicroscope with a built-in Leica MC 170 HD Camera and by SEM after fixing the characters. The standards for euthanizing, fixation, SEM protocols, live specimen photography, and morphological descriptive nomenclature followed the current Neopatida literature (*Monge-Nájera & Morera-Brenes, 1994*; *Costa, Chagas-Junior & Pinto-da-Rocha, 2018*). We focused on the morphological traits of the Neotropical species, especially those with the type-locality nearest to the specimens analyzed here. Here, we adopted the redescription of species based on type series and the new specimens instead of just the holotype (compound description), as the results and illustrations were based on fresh specimens.

**Table 1** **Sequencing data used in the phylogenetic analyses among *Epiperipatus* species.** Samples that were included in the analyses with their respective specimen information, collection data and GenBank accession numbers. GenBank accessions for newly obtained data are underlined.

| Morphotype | ID Number | Voucher | Locality | COI | 12S | 16S | 18S |
|---|---|---|---|---|---|---|---|
| *Epiperipatus beckeri* | MNRJ 0045 | ony_002 | Brazil, Bahia, Camacan | MN905627 | MN639356 | MN544103 | MN705438 |
| *Oroperipatus corradoi* | MNRJ 0071 | ony_005 | Ecuador, Lora, Zamora huayco | MN905629 | MN639357 | MN544105 | MN705439 |
| *Epiperipatus ohausi* | MNRJ 0056 | ony_011 | Brazil, Rio de Janeiro, Nova Iguaçu | MN905634 | MN639361 | MN544109 | MN705441 |
| *Oroperipatus* sp. | MNRJ 0072 | ony_013 | Ecuador, Chimborazo, Sibambe | MN905636 | MN639362 | MN544111 | MN705442 |
| *Oroperipatus* sp. | MNRJ 0069 | ony_014 | Ecuador, Esmeraldas, Chuchuvi | MN933398 | MG973711 | MN933786 | MG973604 |
| *Epiperipatus* sp8 | MNRJ 0055 | ony_015 | Brazil, Pernambuco, Tamandaré | MN905637 | MN639363 | MN544112 | MN705443 |
| *Epiperipatus* sp. | ONY-19 | ony_020 | Brazil, Mato Grosso do Sul, Bodoquena | MN905639 | MN639365 | MN544114 | MN705444 |
| *Epiperipatus acacioi* | UFMG | ony_025 | Brazil, Minas Gerais, Ouro Preto, Tripuí | MN905641 | MN639367 | MN544116 | – |
| *Epiperipatus acacioi* | UFMG | ony_026 | Brazil, Minas Gerais, Ouro Preto, Tripuí and Itacolomi | MN905642 | – | – | – |
| *Epiperipatus puri* | MNRJ 0088 | ony_036 | Brazil, Rio de Janeiro, Cachoeiras de Macacu | MN905648 | MN639375 | MN544124 | MN705445 |
| *Epiperipatus acacioi* | MNRJ 0044 | ony_038 | Brazil, Minas Gerais, Ouro Preto, Tripuí | HQ655618 | HQ404920 | MG973517 | MG973554 |
| *Epiperipatus titanicus* | MNRJ 0053 | ony_043 | Brazil, Alagoas, Murici | MN905652 | MN639378 | MN544128 | MN705446 |
| *Epiperipatus lucerna* | MNRJ 0102 | ony_044 | Brazil, Alagoas, Murici | MN905653 | MN639379 | MN544129 | MN705447 |
| *Epiperipatus hyperbolicus* | MNRJ 0105 | ony_045 | Brazil, Alagoas, Murici | MN905654 | MN639380 | MN544130 | MN705448 |
| *Epiperipatus hyperbolicus* | MNRJ 0104 | ony_046 | Brazil, Alagoas, Murici | MN905655 | MN639381 | MN544131 | MN705449 |
| *Epiperipatus lucerna* | MNRJ 0101 | ony_047 | Brazil, Alagoas, Murici | MN905656 | MN639382 | MN544132 | – |
| *Epiperipatus acacioi* | MNRJ 0044 | ony_054 | Brazil, Minas Gerais, Ouro Preto, Tripuí | MN905661 | MN639385 | MN544138 | MN705450 |
| *Epiperipatus* sp9 | MNRJ 0054 | ony_055 | Brazil, Pernambuco, Caruaru | MN905662 | MN639386 | MN544139 | MN705451 |
| *Epiperipatus* sp9 | MNRJ 0054 | ony_056 | Brazil, Pernambuco, Caruaru | MN905663 | MN639387 | MN544140 | MN705492 |
| *Epiperipatus* sp9 | MNRJ 0054 | ony_057 | Brazil, Pernambuco, Caruaru | MN905664 | MN639388 | MN544141 | MN705452 |
| *Epiperipatus edwardsii* | IZ-141306 | ony_059 | Guiana Francesa, Nouragues Field Station | MN905666 | MN933780 | HG531962 | MG973542 |
| *Epiperipatus biolleyi* | MZUSP 0012 | ony_060a | Costa Rica, San José, Cascajal de Coronado | MN905667 | MN639389 | MN544142 | MN705453 |
| *Mongeperipatus solorzanoi* | MZUSP 0013 | ony_063 | Costa Rica, Limón, Guayacán de Siquirres | MN905670 | MN639393 | MN544147 | MN705454 |
| *Epiperipatus* sp. | ONY-30 | ony_066 | Brazil, Mato Grosso do Sul, Bodoquena | MN905673 | MN639395 | MN544149 | MN705455 |

| Morphotype | ID Number | Voucher | Locality | COI | 12S | 16S | 18S |
|---|---|---|---|---|---|---|---|
| *Epiperipatus* sp. | ONY-32 | ony_067 | Brazil, Mato Grosso do Sul, Bodoquena | MN905674 | MN639396 | MN544150 | MN705456 |
| *Epiperipatus* sp. | ONY-31 | ony_068 | Brazil, Mato Grosso do Sul, Bodoquena | MN905675 | MN639397 | MN544151 | MN705457 |
| *Epiperipatus* sp. | MZUSP 0018 | ony_071 | Brazil, Pará state, Altamira | MH107355 | MG973709 | MG973469 | MG973595 |
| *Epiperipatus bouvieri* | ICN-ONY-27 | ony_074 | Colombia, Cundinamarca, San Antonio del Tequendama | MN905677 | MN639400 | MN544153 | MN705458 |
| *Epiperipatus bouvieri* | ICN ONY-28 | ony_075 | Colombia, Cundinamarca, San Antonio del Tequendama | MN905678 | MN639401 | MN544154 | MN705459 |
| *Epiperipatus* sp. | MZUSP 0021 | ony_077 | Brazil, Pará, Parauapebas | MN905680 | MN639403 | MN544156 | MN705461 |
| *Epiperipatus marajoara* | MZUSP 0022 | ony_078 | Brazil, Pará, Breves | MN905681 | MN639404 | MN544157 | MN705462 |
| *Epiperipatus marajoara* | MZUSP 0023 | ony_079 | Brazil, Pará, Breves | MN905682 | MN639405 | MN544158 | MN705463 |
| *Epiperipatus marajoara* | MZUSP 0024 | ony_080 | Brazil, Pará, Breves | MN905683 | MN639406 | MN544159 | MN705464 |
| *Epiperipatus marajoara* | MZUSP 0025 | ony_081 | Brazil, Pará, Breves | MN905684 | MN639407 | – | MN705465 |
| *Epiperipatus marajoara* | MZUSP 0026 | ony_082 | Brazil, Pará, Breves | MH107345 | MG973705 | MG973529 | MG973543 |
| *Epiperipatus marajoara* | MZUSP 0027 | ony_083 | Brazil, Pará, Breves | MN905685 | MN639408 | MN544160 | MN705466 |
| *Epiperipatus* [sp4] | MZUSP 0028 | ony_084 | Brazil, Pará, Bragança | MN905686 | MN639409 | MN544161 | MN705467 |
| *Epiperipatus* sp. | MZUSP 0019 | ony_094 | Brazil, Goiás, São Domingos, Angélica cave | MH107356 | MG973655 | MG973478 | MG973548 |
| *Epiperipatus* sp. | MZUSP 0029 | ony_096 | Brazil, Bahia, Campo Formoso | MN905693 | MN639414 | MN544167 | MN705468 |
| *Epiperipatus titanicus* | MNRJ 0053 | ony_107 | Brazil, Alagoas, Murici | MN905699 | MN639425 | MN544177 | MN705494 |
| *Epiperipatus lucerna* | MNRJ 0103 | ony_108 | Brazil, Alagoas, Murici | MN905700 | MN639426 | MN544178 | MN705470 |
| *Epiperipatus* sp. | MZUSP 0073 | ony_112 | Colombia, Santander, Pamplona | MN905702 | MN639428 | MN544180 | MN705471 |
| *Epiperipatus* sp. | MZUSP 0072 | ony_113 | Brazil, Amazonas, Silves | MN905703 | MN639429 | – | MN705495 |
| *Epiperipatus* sp. | UFPB PG 0008 | ony_118 | Brazil, Alagoas, Ibateguara, Serra Grande | MN905705 | MN639433 | – | MN705473 |
| *Epiperipatus* sp3 | MZUSP 0090 | ony_130 | Brazil, Ceará, Ubajara | MH107338 | MG973653 | MG973479 | MG973550 |
| *Epiperipatus* sp3 | MZUSP 0089 | ony_131 | Brazil, Ceará, Ubajara | MN905709 | MN639440 | MN544191 | MN705478 |
| *Epiperipatus* sp3 | MZUSP 0091 | ony_132 | Brazil, Ceará, Ubajara | MH107339 | MG973654 | MN933789 | MG973551 |
| *Epiperipatus* sp. | MZUSP 0100 | ony_138 | Brazil, Minas Gerais, Itabirito | MN905710 | MN639446 | – | MN705489 |
| *Epiperipatus vagans* | MZUSP 0101 | ony_139 | Panama, Panama City | MH107349 | MG973663 | MG973482 | MG973544 |
| *Epiperipatus bernali* | MZUSP 0106 | ony_143 | Panama, Chiquiri, Universidade Autonoma de Chiriqui | MH107364 | MG973686 | MG973493 | MG973573 |
| *Epiperipatus bernali* | MZUSP 0107 | ony_144 | Panama, Chiquiri, Universidade Autonoma de Chiriqui | MH107362 | MG973689 | MG973492 | MG973574 |
| *Epiperipatus* sp. | MZUSP 0110 | ony_147 | Panama, Chiquiri, Universidade Autonoma de Chiriqui | MH107363 | MG973688 | MG973494 | MG973579 |
| *Epiperipatus vagans* | MZUSP 0113 | ony_150 | Panamá, Panama City, Parque Nacional Soberanía | MH107348 | MG973664 | MG973483 | MG973545 |
| *Epiperipatus vagans* | MZUSP 0114 | ony_151 | Panama, Panama City, Parque Nacional Soberanía | MH107350 | MG973665 | MG973484 | MG973547 |
| *Epiperipatus vagans* | MZUSP 0115 | ony_152 | Panama, Panama City, Parque Nacional Soberanía | MH107347 | MG973666 | MG973485 | MG973546 |

**Table 1** (*continued*)

| Morphotype | ID Number | Voucher | Locality | COI | 12S | 16S | 18S |
|---|---|---|---|---|---|---|---|
| *Eoperipatus* cf. *horsti* | IZ-131341 | DNA103566 | Malaysia | KC754636 | KC754471 | KC754519 | MG973605 |
| *Peripatopsis lawrencei* | IZ-131348 | DNA103588 | South Africa, Western Cape; Off Franschoek | KC754687 | KC754514 | KC754568 | MG973647 |
| *Peripatopsis moseleyi* | IZ-131346 | DNA103586 | South Africa, Eastern Cape; Keiskammahoek | KC754688 | KC754515 | KC754569 | – |
| *Peripatus basilensis* | IZ-131422 | DNA104977 | Dominican Republic | KC754646 | MG973700 | MG973471 | MG973563 |
| *Mesoperipatus tholloni* | IZ-131381 | DNA104625 | Gabon | KC754645 | KC754478 | KC754528 | KC754576 |
| *Principapillatus hitoyensis* | IZ-131339 | DNA103564 | Costa Rica, Limón | MH107340 | MG973680 | MG973488 | MG973555 |
| *Epiperipatus acacioi* | MNRJ 0119 | ITV50358 | Brazil, Minas Gerais, Nova Lima, cave CPMT_0015 | PP054359 | PP060402 | PP051244 | PP060417 |
| *Epiperipatus acacioi* | MNRJ 0120 | ITV50359 | Brazil, Minas Gerais, Nova Lima, cave CPMT_0008 | PP054360 | PP060403 | PP051245 | PP060418 |
| *Epiperipatus acacioi* | MNRJ 0121 | ITV50360 | Brazil, Minas Gerais, Nova Lima, Mina de Abóboras, cave ABOB_0009 | PP054358 | PP060404 | PP051246 | PP060419 |
| *Epiperipatus acacioi* | MNRJ 0111 | ITV50361 | Brazil, Minas Gerais, Rio Acima, Vale-Abóboras; cave ABOB-0028 | PP054357 | PP060405 | PP051247 | PP060420 |
| *Epiperipatus acacioi* | MNRJ 0112 | ITV50362 | Brazil, Minas Gerais, Rio Acima, Vale-Abóboras; cave ABOB-0040 | PP054361 | PP060406 | PP051248 | PP060421 |
| *Epiperipatus acacioi* | MNRJ 0113 | ITV50363 | Brazil, Minas Gerais, Rio Acima, Vale-Andaime, cave SM_0030 | PP054362 | PP060407 | PP051249 | PP060422 |

## Molecular data acquisition and molecular inferences

We analyzed the molecular dataset of 67 terminals (Table 1) within the framework of total evidence. Phylogenetic inferences were evaluated under probabilistic frameworks based on previously published data from *Giribet et al. (2018)* and new sequences we obtained for six specimens. For the newly analyzed specimens, we performed the genomic DNA extractions with the DNeasy Blood & Tissue kit (Qiagen, Hilden, Germany), following the modifications recommended for insect tissues by the manufacturer. Both the quantity and quality of the isolated DNA were verified with a Qubit 3.0 (Invitrogen, Waltham, MA, USA) fluorimeter using the Qubit dsDNA HS kit (Invitrogen) and a NanoDrop One[C] (Thermo Scientific, Waltham, MA, USA) spectrophotometer, respectively. Then, we used approximately 1–10 ng of the extracted DNAs to prepare paired-end libraries with the Illumina DNA Prep kit (Illumina, San Diego, CA, USA), using the XGEN Nextera adapters (Integrated DNA Technologies, Coralville, IA, USA), following the manufacturer's protocol. The libraries were sequenced with a NextSeq 1000/2000 P2 300 cycles (2 × 150 pb) in a NextSeq 2000 platform. Obtained raw reads were treated with AdapterRemoval v2.3 (*Schubert, Lindgreen & Orlando, 2016*), using PHRED > 20 as a quality threshold. Subsequently, the mitochondrial (COI, 12S rRNA and 16S rRNA) and nuclear (18S rRNA) genes were obtained through *de novo* assemblies with the software NovoPlasty v4.3 (*Dierckxsens, Mardulyn & Smits, 2017*), using sequences of COI (ABF93293) and 18S (MG973570) of *E. biolleyi* (Bouvier, 1902) available in the GenBank (https://www.ncbi.nlm.nih.gov/genbank/) database as seeds. Sequences were checked and corrected with Geneious Prime v2023 (Biomatters, Auckland, New Zealand), aligned with

MAFFT v.7 (*Katoh & Standley, 2013*), and concatenated using SequenceMatrix (*Vaidya, Lohman & Meier, 2011*).

For the phylogenetic reconstruction using the maximum likelihood (ML) approach, we used RAxML v.8.1.11 (*Stamatakis, 2014*), employing a partitioned GTRGAMMA model specifying each partition by gene and further by codon position within COI (positions 1 and 2 comprised a single partition while position 3 constituted its own). Also, we applied node dating in BEAST v.2.6.2 (*Bouckaert et al., 2019*) using nucleotide substitution model averaging, as implemented in bModelTest (*Bouckaert & Drummond, 2017*), a relaxed clock log-normal model and a birth-death model of speciation, which seems to work well with a mix of species and population-level datasets (*Ritchie, Lo & Ho, 2017*), with 100,000,000 generations, sampling one tree out of every 1,000, subsequently applying a burn-in fraction of 10%. For this calibration, we followed *Giribet et al. (2018)*, assigning a normal distribution for the age of the root in Onychophora as the calibration point, with a mean of 298.75 Ma and a standard deviation of 10 (equivalent to 30 Ma), while for the West Gondwanan clade divergence from *Eoperipatus*, we applied a log-normal distribution with the offset at 98 Ma and a standard deviation of 2.2, which allows the 97.5% quartile to be as old as 300 Ma. Posterior probabilities and parameter ESS values were visualized in Tracer 1.7 (*Rambaut et al., 2018*) and used to assess convergence and decide the burn-in fraction, with all ESS values > 200. A summary tree was generated with TreeAnnotator v. 2.6.2 (from the BEAST package) using median heights and a maximum clade credibility tree. The ML and Bayesian analyses were run *via* the CIPRES Science Gateway (*Miller, Pfeiffer & Schwartz, 2010*). Finally, we adopted the dated tree as our working hypothesis, considering that such an approach was the most informative among the two resulting reconstructions.

## RESULTS

### Phylogenetic analysis

Our phylogenetic hypothesis comprised 57 terminals of *Epiperipatus* used in *Giribet et al. (2018)* and *Costa, Giribet & Pinto-da-Rocha (2021)*, including all previously published *E. acacioi* sequences, besides our sequenced data for six newly sampled accessions of this species, as evidenced in Table 1. Recent studies based on a vast molecular dataset have helped to understand the relationships within Neopatida, and *E. brasiliensis*, *E. bouvieri*, *E. acacioi*, and *E. cratensis* appeared nested in the largest clade of Neotropical species referred to as *Epiperipatus* (see *Costa, Giribet & Pinto-da-Rocha, 2021*; Fig. 2 and Fig. S1).

Our strategy to analyze a concatenated dataset of four fragments, in which we used DNA sequencing data from 67 terminals, worked well (Fig. 2). In the ML and Bayesian analyses, we recovered the sister group relationship between Peripatopsidade and Peripatidae (Fig. 2, clades A and B), with the latter being divided into Southeast Asian *Eoperipatus*, with the central African species *Mesoperipatus tholloni* (*Bouvier, 1898*) as sister to Neopatida (clade C). We also observed the *Oroperipatus* terminals within the Neopatida clade, nested in a clade that is sister to the rest of the Caribbean peripatids (clade D). In this latter clade, the Brazilian specimens were distributed into six clades as follows: clade E with specimens

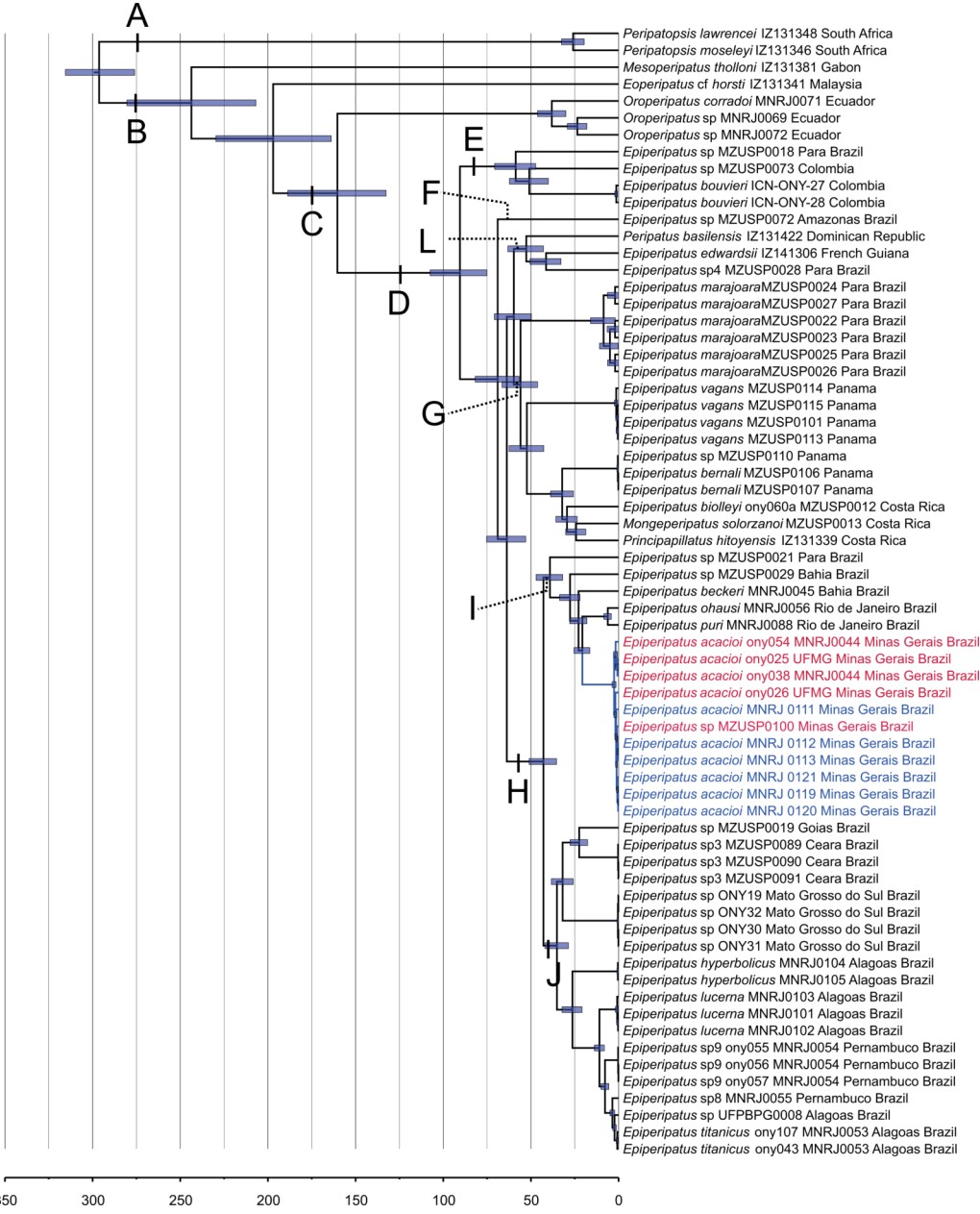

**Figure 2 Phylogenetic relationships among *Epiperipatus* species.** BEAST chronogram without outgroups obtained by *Giribet et al. (2018)*, showing divergence time estimates for Peripatidae, with updates to the list of terminals. The terminals in red (type locality) and blue (Rio Acima) indicate the *Epiperipatus acacioi* clade. Note the close relationship among the sequences.

from Colombia; clade F with a single terminal from an Amazonian specimen; clade L with taxa from the Dominican Republic and French Guiana; clade G with species from Panama and Costa Rica; and the exclusively Brazilian clade H. Despite the position of *E. edwardsii* (*Blanchard, 1847*) in the clade L of the chronogram, we considered clade D as the *Epiperipatus* group, following *Giribet et al. (2018)* and *Costa, Giribet & Pinto-da-Rocha (2021)*, as our dataset did not encompass the entire diversity of Neotropical onychophorans. We have also performed an analysis with a dataset with 64 terminals with no missing data, and the obtained relationships were the same as those presented in the chronogram of Fig. 2.

In a closer look at the Brazilian clade, it is evident that the new specimens from Nova Lima and Rio Acima are closely related to the previously analyzed specimen of *E. acacioi*, forming a clade with other species such as *Epiperipatus* sp. (from Bahia), *E. ohausi* (Bouvier, 1900), *E. puri Costa, Mendes & Giupponi, 2023*, and *E. beckeri Costa, Chagas-Junior & Pinto-da-Rocha, 2018* (clade I). Based on this relationship, we have concluded that the new specimens belong to a previously unknown population of *E. acacioi*, thus extending its occurrence beyond the type locality. Based on our molecular phylogenetic reconstructions, we provide a redescription of *E. acacioi* based on specimens from Rio Acima collected in 2019.

## Morphology

The characters illustrated here include dorsal papillae, jaw, fourth leg in ventral view, and head in anteroventral view obtained under SEM and a stereomicroscope.

## Systematics

Family **Peripatidae** *Audouin & Milne-Edwards, 1832*.
Genus ***Epiperipatus*** *Clark, 1913*.

*Epiperipatus acacioi* (*Marcus & Marcus, 1955*)
*Peripatus acacioi Marcus & Marcus, 1955*: 189
*Peripatus ouropretanus* Trindade, 1958: 520
*Peripatus* (*Macroperipatus*) *acacioi*: *Froehlich, 1968*: 168; *Castro & Silva, 2001*: 1035 (misidentification)
*Macroperipatus acacioi*: *Peck, 1975*: 346; *Sampaio-Costa, Chagas-Junior & Baptista, 2009*: 557 *Epiperipatus acacioi*: *Oliveira, Wieloch & Mayer, 2010*: 21; *Oliveira, Read & Mayer, 2012*: 7; *Oliveira, 2023*: 142

(Figs. 3–5, Figs. S2 and S3)

Type material: Type material. **MZUSP 0048**: ♂ (lectotype designated here) 26 leg pairs, 2 ♀, 1 ♂ (paralectotypes) and 3 embryos, BRAZIL, Minas Gerais, Ouro Preto, no date data, Acácio Costa leg.

Examined material. **MNRJ 0044**, 21 ♀, 3 ♂, BRAZIL, Minas Gerais, Ouro Preto, Estação Ecológica do Tripuí, 11.vii.2009, Costa, C. S. and Oliveira, I de S. leg. **DZUFMG-ONY0041**,

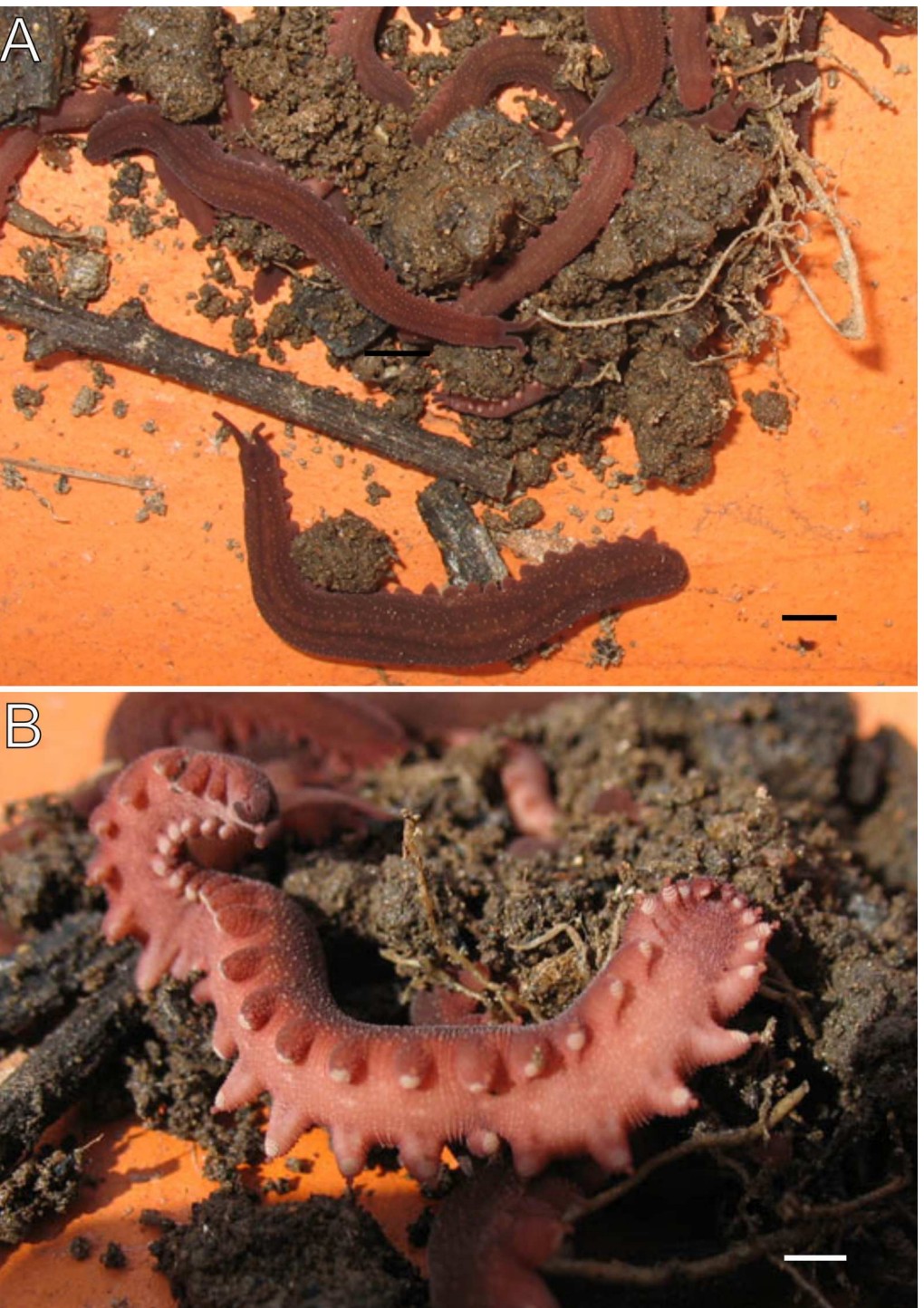

**Figure 3  Habitus and body color of *Epiperipatus acacioi*.** Body background of *E acacioi*, MNRJ 0044, Ouro Preto. (A) Dorsal side. (B) Ventral side. Note the ventral organs main body axis. Scales bars in A = 4 mm; and B = 2 mm.

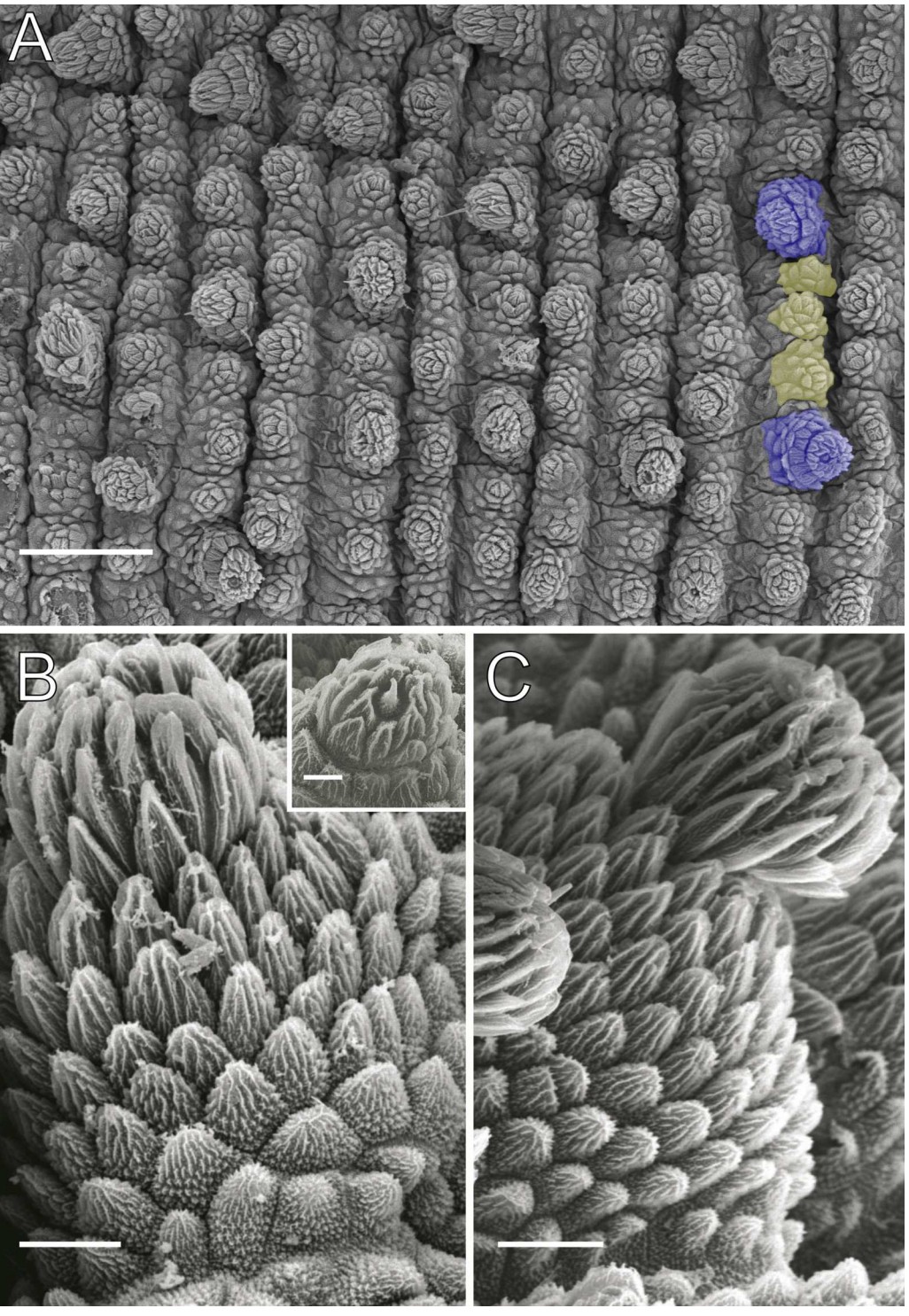

**Figure 4 Dorsal body details of *Epiperipatus acacioi*.** Scanning electron micrographs of the body of *E. acacioi*, MNRJ 0044, Ouro Preto. (A) Dorsal papillae arrangement near to the dorsomedian furrow, with the detail of the primary and accessory papillae highlighted in blue and yellow, respectively. (B) Primary papillae posterior region. Note the small box, detailing another apical piece with one scale in the same view. (C) Primary papillae in anterior view. Scales bars in A = 100 μm; B and C = 20 μm; and detail in the upper right corner in B = 10 μm.

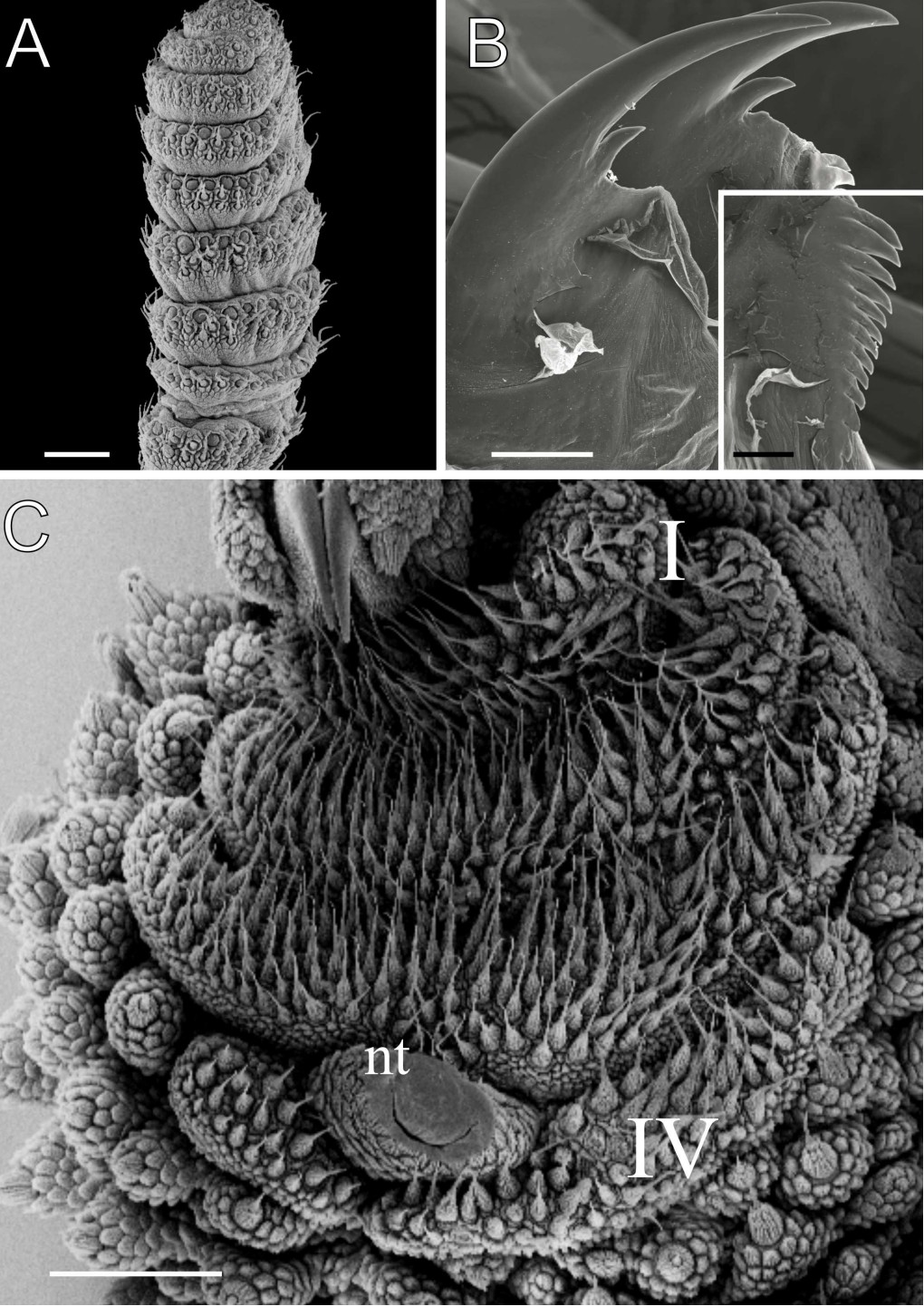

**Figure 5** **Head and oncopod parts details of *Epiperipatus acacioi*.** Scanning electron micrographs of the head of *E. acacioi*, MNRJ 0044, Ouro Preto. (A) Right antenna. Note the sequence of seven large rings followed by narrow rings at the top. (B) Right jaw (outer blade in the foreground). Detail of the series of denticles on the outer blade in the small box. (C) Spinous pad of the fourth right leg. The Roman numbers indicate the first and last spinous pads. Legend: nt = nephridial tubercle. Scales bars in A–C = 100 μm; and detail in the lower right corner in B = 50 μm.

1 ♂, idem, 10.iv.2008, Oliveira, I de S. leg. **DZUFMG-ONY0049**, 1 ♂, idem, 15.v.2008, idem. **DZUFMG-ONY 0051**, **0052**, **0054**, **0056**, **0158**, 5 ♂, Parque Estadual do Itacolomi, 05-06.vi.2008, idem. **DZUFMG-ONY 0164**, **0166**, **0167**, 3 ♂, idem, 11-13.x.2008, idem. **MNRJ 0111**, 1 ♀, Rio Acima, Vale S/A (Mina de Abóboras), cave ABOB_0028, 13.vi.2019, Spelayon and collaborators. **MNRJ 0112**, unsexed specimen, Itabirito, Vale S/A (Mina de Abóboras), cave ABOB_0040, 29.iii.2019, idem. **MNRJ 0113**, 1 ♂, Itabirito, Vale S/A (Andaime), cave SM_0030, 29.vii a 02.viii.2019, idem. **MZUSP 0100**, unsexed, Itabirito, cave VL-32, 07-10.v.2013, Mascarenhas, J. and collaborators leg. **MNRJ 0119**, unsexed specimen, Nova Lima, Vale S/A (Mina Horizontes), cave CPTM_0015, vi.2016, Robson de Almeida Zampaulo leg. **MNRJ 0120**; unsexed specimen, Nova Lima, Vale S/A (Mina Horizontes), cave CPTM_0008, 25.ii.2017, Robson de Almeida Zampaulo leg. **MNRJ 0121**, 1 unsexed specimen, Nova Lima, Vale S/A (Mina de Abóboras), cave ABOB_0009, 25.ii.2017, Robson de Almeida Zampaulo leg.

Emended diagnosis. Dorsal body background dark red, with twelve uniformly wide plicae, including incomplete ones. Dorsal papillae fit smoothly on the plicae. Primary papillae have roundish basal pieces and regular spherical apical piece. Accessory papillae are shorter than primary papillae. Number of legs ranges from 24 to 30 pairs.

Redescription. Measurements. Males: length 13–30 mm; width 2.0–3.0 mm; height 1.0–2.0 mm. Females (slightly larger than males): length 18–51 mm; width 2.0–5.0 mm; height 1.0–3.0 mm. Syntypes: female length 30 mm; male length 25 mm. Color (living specimens). Dorsal body background dark red, overlaid by a wavy, blurry light yellowish-pink band extending along the longitudinal axis (Fig. 3A). Dorsomedian furrow dark purplish red, overlapped by diamond-shaped marks matching the dorsal background color (Fig. 3A). Antennae and head match the dorsal body color. Ventral surface uniformly red (Fig. 3B).

Body description. Dorsomedian furrow clearly evident and hyaline organs are inconspicuous (Fig. 3A). Each segment has 12 plicae, including incomplete plicae. Seven complete plicae cross to the ventral side, bearing dorsal papillae (Fig. 4A; Fig. S2). Two primary papillae separated by rows of accessory papillae. The number of accessory papillae varies widely, sometimes forming sequences of more than five between two primary papillae (Fig. 4A). The largest primary papillae are on alternate dorsal plicae and never on incomplete plicae (Fig. 4A). Accessory papillae are more abundant than primary papillae on the plicae. Dorsal papillae are positioned on the tops of the plicae, with accessory papillae on the flanks. Both types of dorsal papillae have conical basal pieces. The primary papillae feature an asymmetrical, spherical apical piece (Figs. 4B and 4C; Figs. S2 and S3). Primary papillae are larger than accessory papillae, with the largest ones appearing faded. Additionally, the conical basal pieces of both dorsal papillae are composed of scales that do not overlap at the base. At the basal region of the primary papillae, there are four to nine scale ranks (Figs. 4B–4C). The apical piece consists of overlapping lanceolate scale ranks, three or four in the anterior region and one to three in the posterior region (Figs. 4B and 4C). These scales do not obscure the constriction between the two parts of the primary papillae (Fig. 4A). The bristle is positioned in the posterior region of the apical piece (Fig. 4B).

Head. Typical onychophoran head, featuring eyes, antennae, and slime papillae, with no eversible structures, modified papillae, clefts or furrows. Additionally, no color variation is observed. Antennae consist of 34–45 rings in both sexes, excluding the terminal disc. The antennal tip comprises seven broad rings, followed by an alternating sequence of narrow and broad rings extending nearly to the last antennal rings (Fig. 5A). Eyes and frontal organs are located in the external ventrolateral region of the antennal base (Fig. S3). The frontal organs are equivalent in length to two to four fused antennal papillae. The mouth opening is surrounded by a single small lobe, followed by seven flanking lobes that decrease in size from the anterior to the posterior sides of the mouth. The jaws consist of two blades, each with one long, curved main tooth and one accessory tooth. The dental formula for the inner and outer jaw blades is 1/0–1 and 1/1/8–13, respectively (Fig. 5B; Fig. S3).

Leg pairs: 25–28. The fourth and fifth leg pairs bear at least four complete spinous pads, and a vestigial fifth spinous pad may rarely be present. The nephridial tubercle is located on the fourth and fifth oncopod pairs, positioned between the third and fourth spinous pads, and connected only to the third spinous pad (Fig. 5C; Fig. S3). The fourth spinal pad is deeply concave on the prolateral side, with no clear evidence of a fifth spinous pad (Fig. 5C). Two prolateral and one retrolateral foot papillae are present on the feet of the fourth and fifth legs. The ventral and preventral organs are distinct and resemble those found in other species. The gonopore is located near the penultimate leg pair in both sexes, as in all species of Peripatidae.

Sexual dimorphism. Two or three pregenital legs present, and one or two crural papillae may be present on each. The lectotype exhibits two pregenital legs, each bearing one crural papilla. Inconspicuous anal glands represented only by the respective two pores on the anterior border of anal aperture.

Remarks. Variation of leg pairs: males 24–28; females 26–30. Consequently, the range of leg pairs in this species is extended from 24 to 30, overlapping between sexes.

Distribution. Brazil, Minas Gerais, Estação Ecológica do Tripuí (type locality), and Parque Estadual do Itacolomi, both in Ouro Preto, and Serra da Moeda, which covers the municipalities of Itabirito, Nova Lima, and Rio Acima.

*Epiperipatus brasiliensis* (*Bouvier, 1899*)
(Figs. 6–8)

(*Peripatus brasiliensis Bouvier, 1899*: 1031
*Peripatus* sp. (Peripatus from Amazons): *Moseley, 1879*: 265
*Peripatus* Santarem: *Sedgwick, 1888*: 484 (invalid name)
*Peripatus* (*Epiperipatus*) *brasiliensis*: *Clark, 1913*: 18.
*Epiperipatus brasiliensis*: *Peck, 1975*: 345; *Oliveira, Read & Mayer, 2012*: 9; *Oliveira, 2023*: 145

Examined material. **NHM**, 1 ♀, BRAZIL, Pará, Santarém, no date data; Wickham, purchased from W. H. J. Carter. **NHM 96.5.14-25**, 2 ♀ and 1 ♂, no further data, no collector

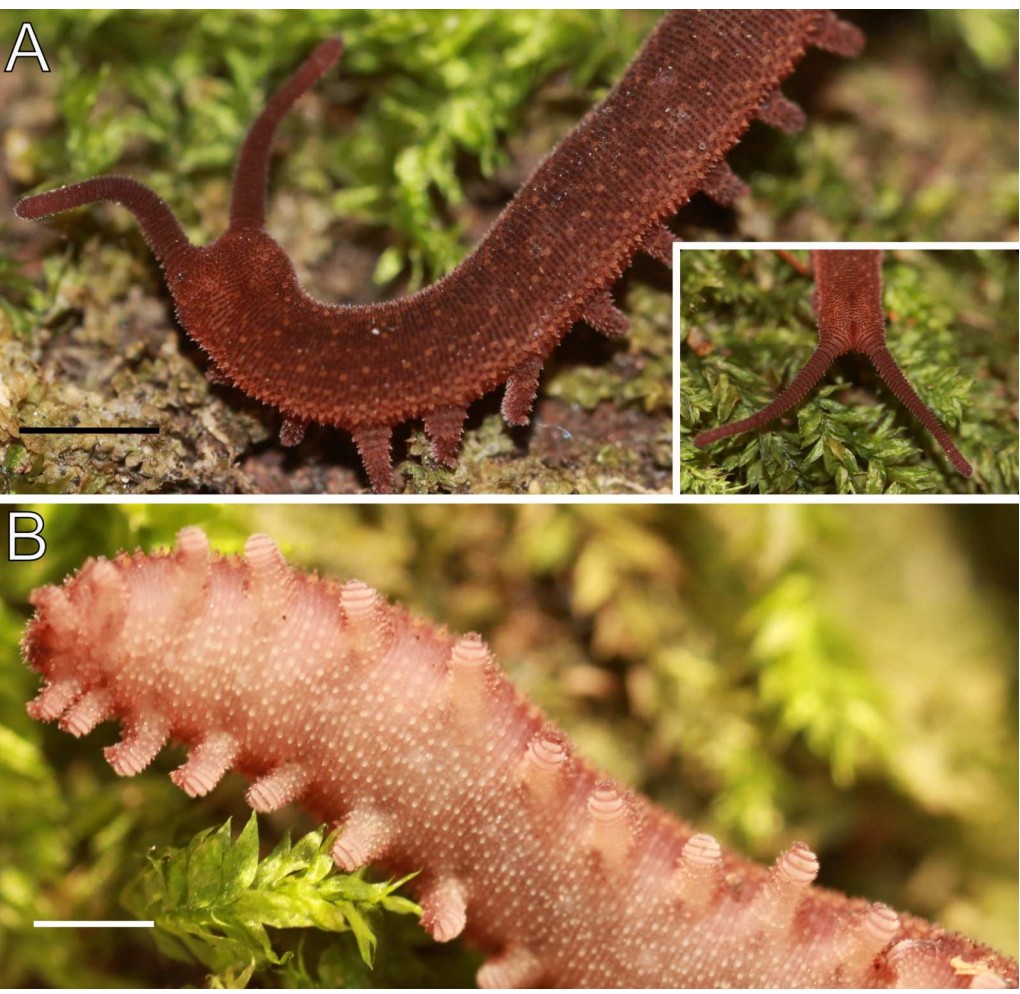

**Figure 6** **Habitus and body color of** *Epiperipatus brasiliensis.* *Epiperipatus brasiliensis*, MZUSP 0121, Santarém, FLONA do Tapajós. (A) Body background of dorsal side. The small box on the right corner shows the frontal side with the small diamond along the main head axis. (B) Body background of ventral side. Scales bars in A and B = 1 mm.

data. **MUZSP 0121**, 1 ♂, Floresta Nacional (FLONA) do Tapajós, km 84, 30.xii.2014, Costa, C.S., Cabra García, J.J., Chirivi, D. and Coronato, A. leg. Type material not examined.

Emended diagnosis. Dorsal body background reddish-brown, with darker dorsomedian furrow originating as a brown spot between antennae insertions. Dorsal plicae extend near the legs, with no incomplete plicae. Primary papillae with small roundish basal pieces and cylindrical apical pieces. Number of legs ranges from 29 to 33 pairs.

Redescription. Measurements. Length 22.1 mm; width 1.5 mm; height 1.2 mm. Dorsal body background moderate brown with light brown primary papillae (Fig. 6A). Color (living specimen) Dorsomedian furrow deep brown, with no evident dorsal color patterns (Fig. 6A). The furrow begins at the head as a strong reddish-brown spot resembling small

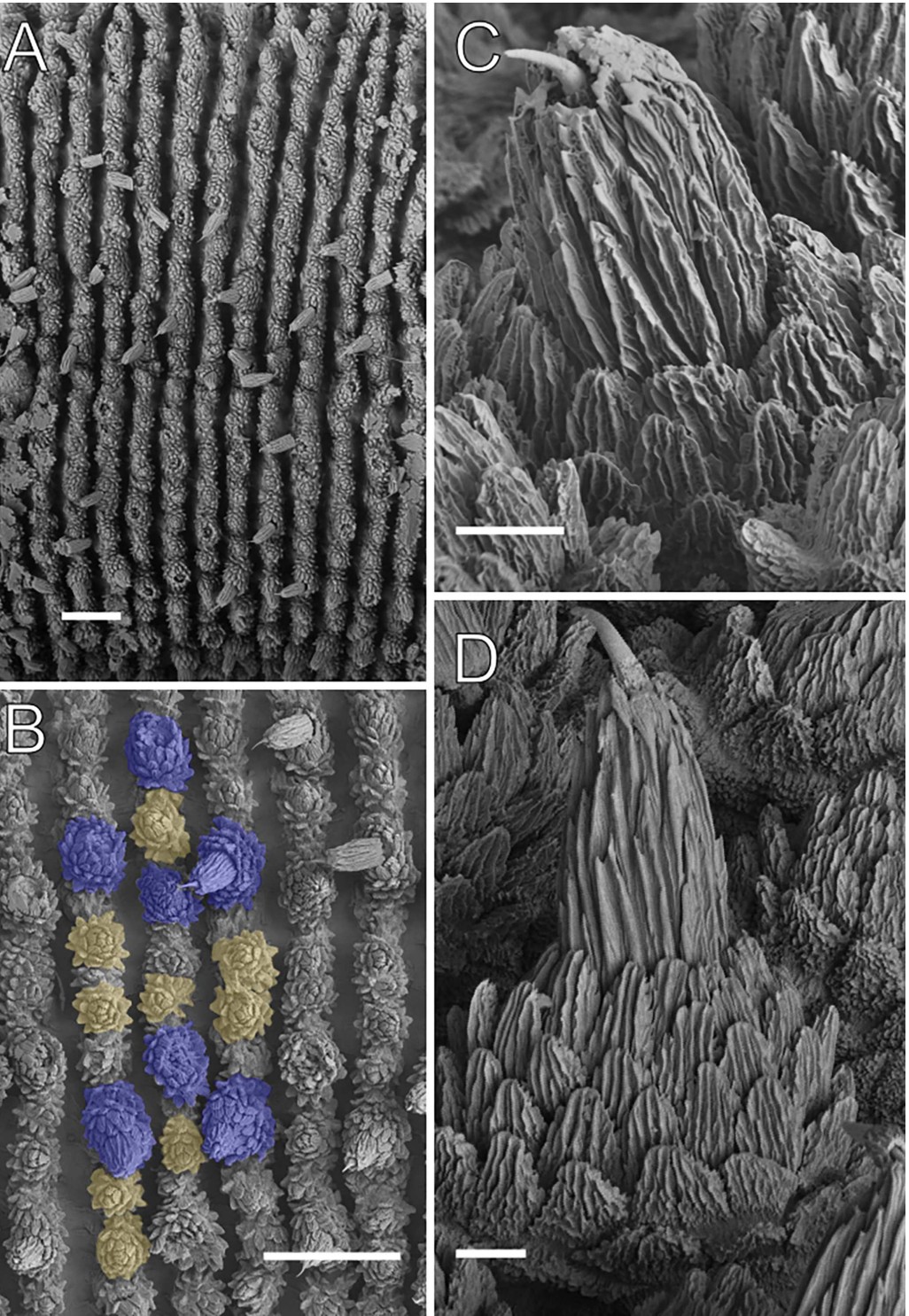

**Figure 7 Dorsal body details of *Epiperipatus brasiliensis*.** Scanning electron micrographs of the body of *E. brasiliensis*, MZUSP 0121, Santarém, FLONA do Tapajós. (A) Dorsal plicae arrangement. 

**Figure 7 (…continued)**
The image shows a complete region from the dorsal (close to the dorsomedian furrow) to ventral (the leg insertion) sides. No evidence of incomplete folds was observed. (B) Detail of the primary papillae (marked in blue) and accessory papillae (highlighted in yellow). Note the primary papillae are bigger than the accessory papillae. (C) Detail of the apical piece in posterior view. (D) Apical piece on anterior view. Note the number of scale ranks is the same as the posterior view. Scales bars in A and B = 100 μm; and C and D = 10 μm.

diamond shapes along the main axis (Fig. 6A). Head and antennae are moderately reddish-brown. Ventral body background moderately reddish-orange, with pale yellowish-orange papillae. Ventral organs light brown, and spinous pads pale yellowish-pink (Fig. 6B).

Body description. Dorsomedian furrow conspicuous, with hyaline organs visible. Each segment features 12 complete dorsal plicae, uniformly wide, with seven extending to the ventral side (Figs. 7A and 7B). All dorsal papillae on the top of the plicae (Fig. 7A). Dorsal papillae with a conical (dome-shaped) basal piece composed of scales overlapping each other at the whole basal piece (Figs. 7B–7D). Primary papillae are the largest dorsal papillae, featuring a round basal dome and asymmetrical conical to cylindrical apical piece (Figs. 7B–7D). Basal piece contains four to five scale ranks, while the apical piece has three anterior and two posterior scale ranks (Figs. 7C and 7D). Needle-shaped sensory bristle is posteriorly directed (Fig. 7C). Accessory papillae, the smallest dorsal papillae, are more numerous and distributed similarly to the primary papillae (Fig. 7B).

Head. No distinct structures or patterns are evident on the head, Except fora black, diamond-shaped marking in the frontal position along the main head axis. Antennae consist of 40–45 rings. The antennal tip comprises seven broad rings (excluding the terminal disc), followed by an alternating sequence of narrow and broad rings, at least up to the 18th ring. Eyes and frontal organs are located in the ventrolateral region of the antennal base. The frontal organs are conspicuous and as long as four fused antennal papillae. The mouth opening is surrounded by a single small anterior lobe and seven flanking lobes, which decrease in size from the anterior to the posterior end of the mouth. The dental formula for the inner and outer jaws is 1/1 and 1/1/9–11, respectively (see *Bouvier, 1905*: 272).

Legs pairs: 29–32. Fourth and fifth pairs bear two prolateral and one retrolateral foot papillae, as observed in other *Epiperipatus* species (Fig. 8A). Nephridial tubercle located on fourth and fifth pairs of legs, between the third and the fourth spinous pads, connected dorsally the third pad (Fig. 8B). Each leg contains four complete spinous pads, with no evidence of a fifth (Fig. 8B). Conspicuous ventral and preventral organs.

Sexual dimorphism. Males exhibit one or two pairs of pregenital legs, each bearing a single crural papilla each, absent in females. Male anal glands are inconspicuous, represented by two pores on the anterior margin of the anal aperture, also absent in females.

Remarks. *Epiperipatus brasiliensis* was the first Onychophoran species described for Brazil by *Bouvier (1899)*, originally recorded in Santarém, Pará, and later reported in Maranhão and Ceará (*Sampaio-Costa, Chagas-Junior & Baptista, 2009*: 556). However, phylogenetic evidence (*Costa, Giribet & Pinto-da-Rocha, 2021*) suggests its distribution, may be restricted to Santarém and nearby areas. Consequently, *E. vagans* Brues, 1925,

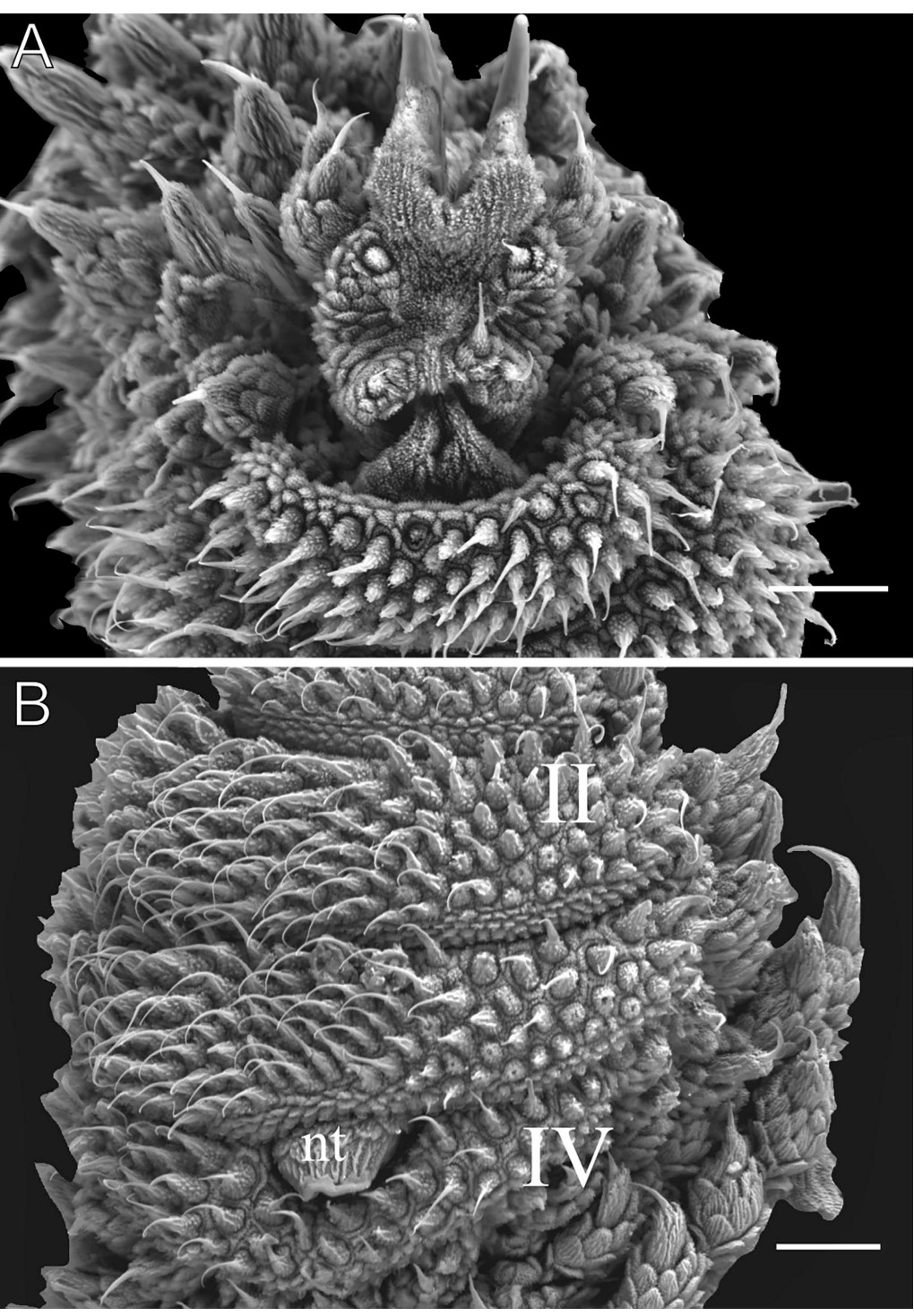

**Figure 8** **Oncopod details of *Epiperipatus brasiliensis*.** Scanning electron micrographs of the head of *E. brasiliensis*, MZUSP 0121, Santarém, FLONA do Tapajós, scanning electron micrographs. (A) Ventral view of the foot of the fourth left oncopod, with two prolateral foot papillae (left side) and one retrolateral foot papilla (right side). (B) The same oncopod as shown in the image above, with a focus on the spinous pads. The Roman numbers indicate the first and last spinous pads. Note the complete fourth spinous pad. Legend: nt = nephridial tubercle. Scales bars in A and B = 40 μm.

which is morphologically similar to *E. brasiliensis* as observed by *Oliveira, Read & Mayer (2012)*, could be regarded as a distinct species. Indeed, *E. brasiliensis* differs from *E. vagans* by some characteristics such as the number of antennal rings and body length, with *E. brasiliensis* being smaller and presenting fewer antennal rings than *E. vagans*. Reports from Maranhão and Ceará likely represent new species or misidentifications, respectively (see the *E. cratensis* description below and *Brito et al., 2010*). In addition, we did not analyze other literature records mentioning *E. brasiliensis* belong to a new species.

Type locality. Santarém, Pará, Brazil.

Distribution. Santarém, Brazil.

*Epiperipatus cratensis* Brito, Pereira, Ferreira, Vasconscellos & Almeida, 2010 (Figs. 9–11)

*Epiperipatus cratensis* *Brito et al., 2010*: 49; *Oliveira, Read & Mayer, 2012*: 9; *Oliveira, 2023*: 146
*Epiperipatus brasiliensis*: *Sampaio-Costa, Chagas-Junior & Baptista, 2009*: 556 (misidentification)

Type material examined. *Epiperipatus cratensis*: **LZ-URCA 701**; 1 ♂ (holotype). **LZ-URCA 591**; 1 ♀, 702; 1 ♂, 703; 1 ♂ (paratypes); BRAZIL, Crato, Ceará, northeastern Brazil; partially lost.

Examined material. **MZUSP 0083**, 1 ♂, BRAZIL, Ceará, Crato, Área de Proteção Ambiental (APA) da Chapada do Araripe, 20-30.i.2014, Sampaio, C. S., DaSilva, M. B. e Saraiva, N. E. V. leg. **MZUSP 0084**. 2 ♀, 1 ♂, FLONA do Araripe-Apodi, 20-30.i.2014, Sampaio, C. S., DaSilva, M. B. e Saraiva, N. E. V. leg. **MNRJ 0022**, unsexed, no further data, Rolim Alencar leg.

Emended diagnosis. Dorsal body background moderate brown, without dorsal color patterns. Twelve complete plicae per segment. Dorsal papillae with a roundish dome-shaped basal piece. Primary papillae with a well-developed, asymmetrical spherical apical piece. Number of legs ranges from 30 to 34 pairs.

Redescription. Measurements. Length 27–34 mm; width 4.7–5.2 mm; height 2.7 mm. Color (living specimens). Dorsal body background moderate brown. Dorsomedian furrow dark grayish-reddish brown, with large light brown primary papillae randomly distributed across the dorsal surface (Fig. 9A). No other dorsal patterns observed. Head and antennae moderately brown (Fig. 9A). Ventral surface brownish orange, with randomly distributed light brown papillae (Figs. 9B and 9C). Ventral organs light grayish-yellowish brown, and spinous pads light yellowish brown (Figs. 9B and 9C).

Body description. Conspicuous dorsomedian furrow and hyaline organs along the main body axis (Fig. 10A). Twelve complete plicae per segment, similar to *E. brasiliensis*. Dorsal plicae uniformly wide (Fig. 10A), with seven extending to the ventral side. Almost all dorsal papillae are situated on the plicae, except for the small accessory papillae located on the flanks of the plicae (Figs. 10A and 10B). Each dorsal papilla consists of a conical,

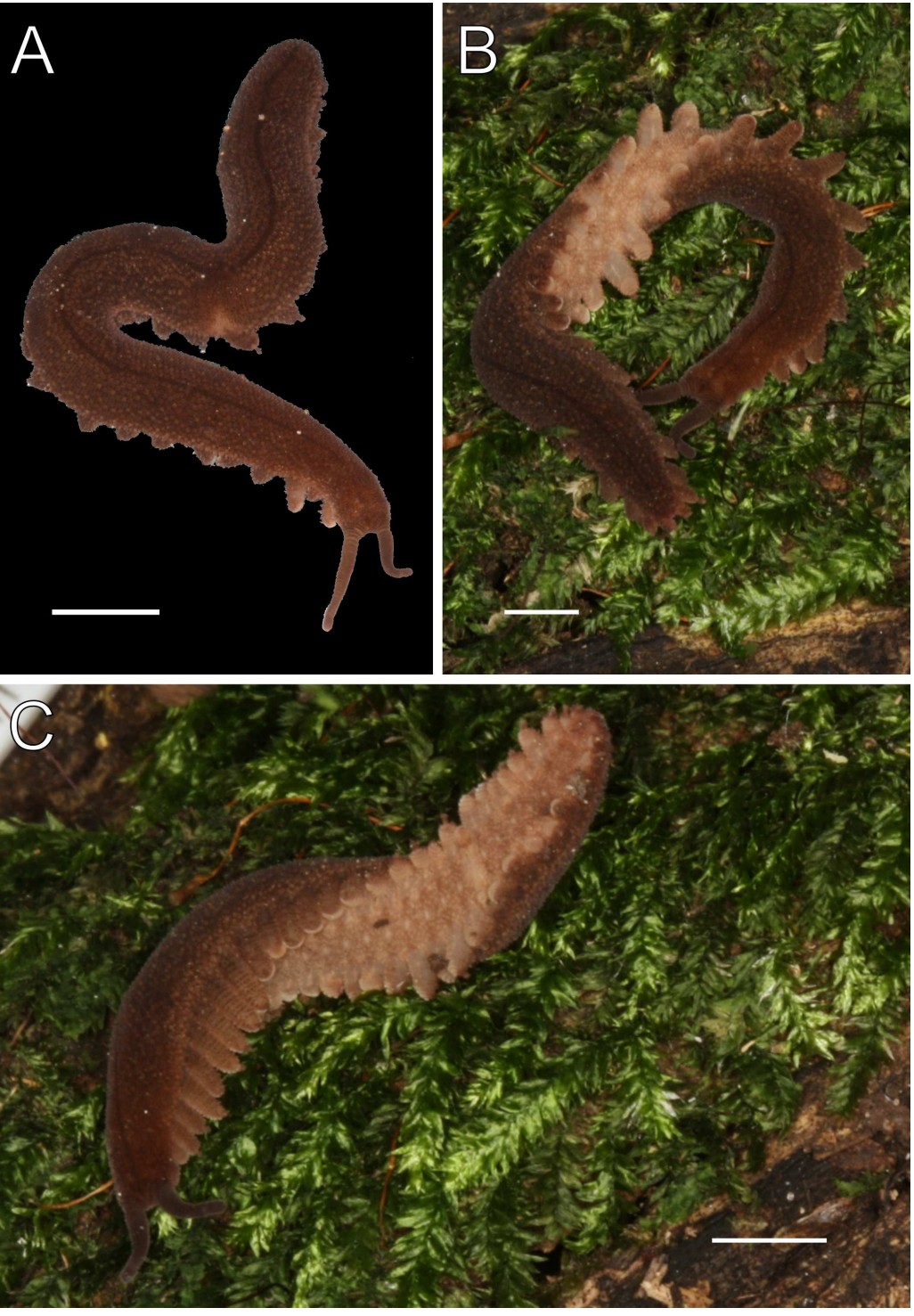

**Figure 9 Habitus and body color of *Epiperipatus cratensis*.** (A) Dorsal view of the body background of *E. cratensis*, MZUSP 0084, Crato. (B and C) Body background of the dorsal and ventral side. The ventral organs are clearly visible. Scales bars in A–C = 4 mm.

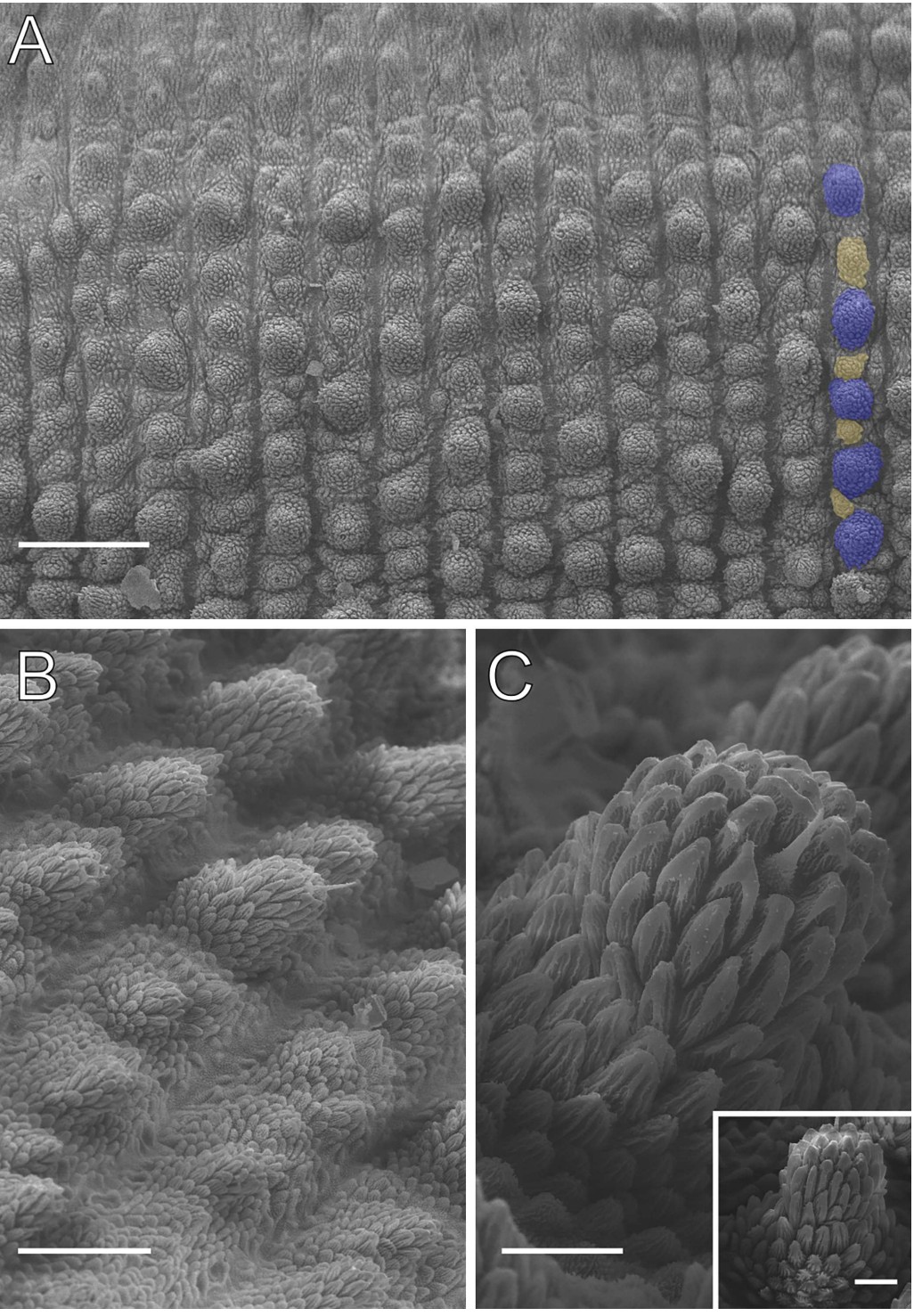

**Figure 10 Dorsal body details of *Epiperipatus cratensis*.** Scanning electron micrographs of the body of *E. cratensis*, Crato. (A) Arrangement of dorsal papillae around the dorsomedial furrow (horizontally at the top of the image), with the detail of the primary and accessory papillae highlighted in blue and in yellow, respectively. (B) Dorsal papillae arrangement in lateral view with the sensory bristle directed posteriorly. 

**Figure 10 (…continued)**
Note the size difference between the primary and accessory papillae as well. (C) Shape of primary papillae in the posterior region. Note the presence of three scale ranks on the apical piece. The anterior region presents a range of four scale ranks as demonstrated in the small box. Scales bars in A = 200 μm; B = 80 μm; and both bars in C = 20 μm.

dome-shaped basal piece, which is entirely covered by overlapping scales (Figs. 10B and 10C). Primary papillae are the largest dorsal papillae, with a round dome-shaped basal piece and a well-developed asymmetrical spherical apical piece (Fig. 10C). However, some primary papillae are as small as the accessory papillae (Fig. 10B). Basal and apical pieces are similar in size. The basal piece contains 5–8 scale ranks, while the apical piece contains 3–6 anterior and 2–3 posterior scale ranks (Fig. 10C). Posteriorly directed needle-shaped sensory bristle present (Figs. 10B and 10C). Accessory papillae, the smallest dorsal papillae, are more abundant and similarly distributed but also occur on the plicae flanks (Fig. 10A). The primary papillae are generally separated by one to five accessory papillae, though they are rarely positioned close together (Fig. 10A). In arrangements with more than three accessory papillae, these are distributed along the tops and flanks of the plicae (Fig. 10A). The dorsal papillae are more widely spaced near the dorsomedian furrow but become progressively closer together toward the ventral side. Only accessory papillae are present on the flanks, though they are rare in this region.

Head. No distinct structures or patterns are evident on the head. The original description reported 31–36 antennal rings, but 41–44 antennal rings were observed in the examined specimens. The antennal tip consists of seven large rings (excluding the terminal disc), followed by alternating narrow and broad rings, at least up to the 25th ring. Eyes and frontal organs are located in the external ventrolateral region of the antennal base. The frontal organs are as long as five fused antennal papillae. The original description noted only five pairs of mouth lobes, but this count appears unlikely. The mouth opening is surrounded by a single small lobe and seven flanking lobes, which decrease in size from the anterior to the posterior end of the mouth. The dental formula for the inner and outer jaws is 1/1 and 1/1/9–10, respectively (Figs. 11A and 11B).

Legs pairs: 33–34 (see *Brito et al., 2010*: 51). Nephridial tubercle located on the fourth and fifth pairs of legs, between the third and fourth spinous pads, connected dorsally to the third spinous pad (Fig. 11C). Two prolateral and one retrolateral foot papillae are present on the feet of the fourth and fifth legs. Each of these legs bears four complete spinous pads, with no evidence of a fifth pad (Fig. 11C). Ventral and preventral organs are evident.

Sexual dimorphism. Males are generally equal to or smaller than females, with some size overlap. Measurements: males—length 41.6–44.0 mm, width 2.1–4.0 mm; females—length 47.4–55.3, width 3.0–3.7 mm. Males have two or three pregenital legs, each bearing one or two crural papillae each, which are absent in females. Male anal glands inconspicuous, represented by two pores on the anterior anal margin, absent in females.

Remarks. Leg pairs variation. Males 30–33; females 33–34. *Sampaio-Costa, Chagas-Junior & Baptista (2009)*: 556) extended the distribution of *E. brasilensis* to include records from Crato. Mello-Leitão identified a specimen from Crato as *Epiperipatus brasiliensis*

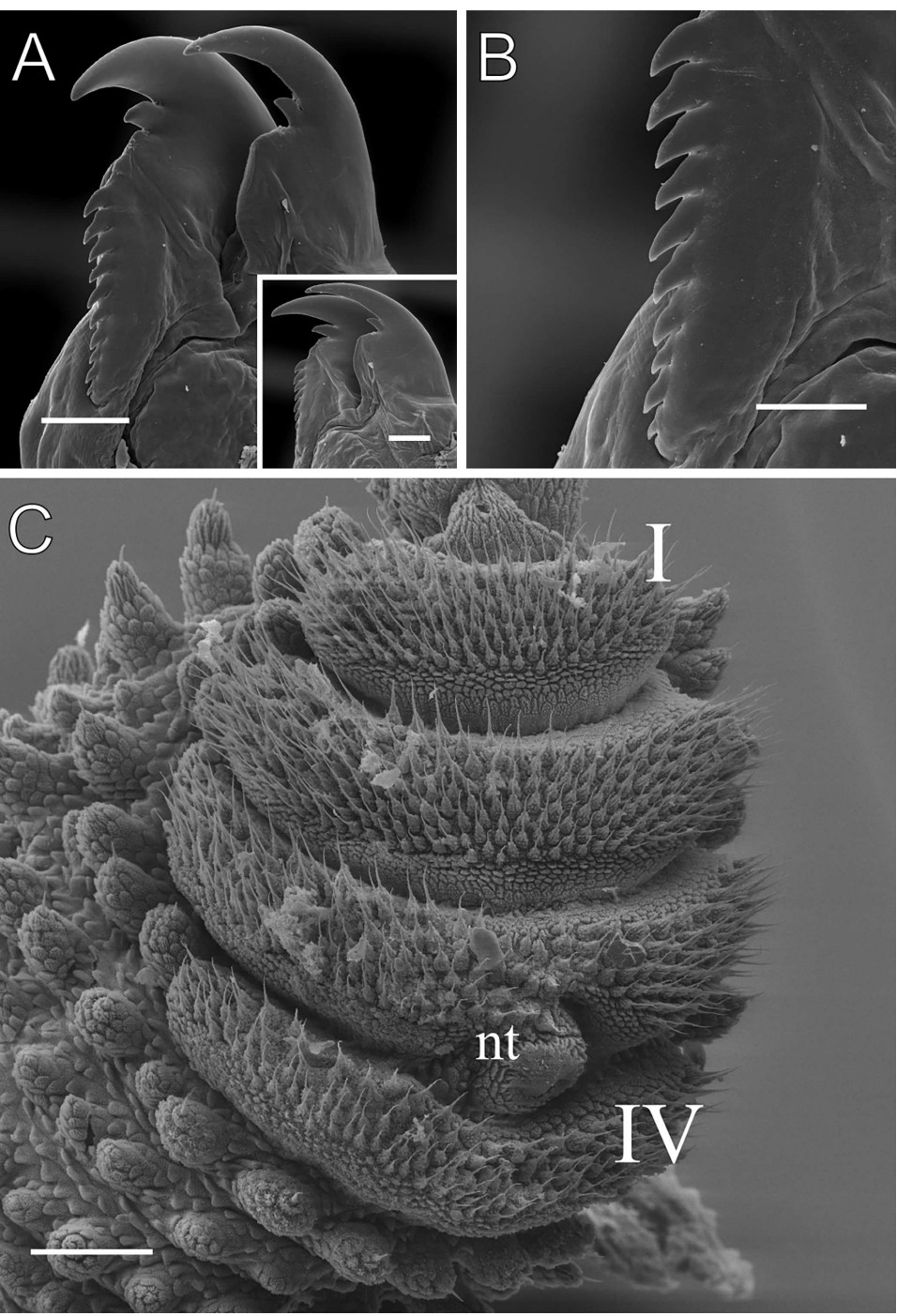

**Figure 11 Head and oncopod parts details of *Epiperipatus cratensis*.** Scanning electron micrographs of the head of *E. cratensis*, MZUSP 0084, Crato. Dental formula and shape. (A) Left jaw (outer blade in first plan). The principal teeth shape is evidenced in the small box. (B) Detail of the denticle series on the outer blade, with ten denticles. (C) Spinous pad of the fourth right leg. The Roman numbers indicate the first and last spinous pads. Legend: nt = nephridial tubercle. Scales bars in A and C = 100 μm; and B = 50 μm.

*xerophilus*, later confirmed by *Sampaio-Costa, Chagas-Junior & Baptista (2009)*. However, *Brito et al. (2010)* did not examine this specimen, suggesting Crato would be a locality shared by both species. The type specimen of *E. cratensis* was originally deposited in the collection of the Universidade Regional do Cariri (URCA) in Crato, under the collection numbers LZ-URCA 701 (holotype), LZ-URCA 591, LZ-URCA 702, and LZ-URCA 703. However, the URCA collection no longer exists, and the holotype has been lost. At URCA, four unidentified specimens were used in the original species description. Two of these specimens bear labels with the following data: 1. Brasil, Ceará, Crato, Nascente do Rio Batateiras, 05.ii.2003, L. A. Souza, *col.* 2. Brasil, Ceará, Crato, Rio Batateiras, 05.viii.2007, Brito, S.V., *col.* Upon examination, these specimens appear to belong to *E. cratensis*. Three specimens from the type series were analyzed in this study and are currently stored in the collection of Prof. Dr. Alexandre Vasconcellos at the Federal University of Paraíba, with no indication of which specimen is the holotype. In addition, *E. cratensis* differs from *E. brasiliensis* in having primary papillae with spherical apical pieces, whereas *E. brasiliensis* has primary papillae with cylindrical apical pieces. Therefore, we conclude that *E. cratensis* is the only species occurring in Crato, likely restricted to the Chapada do Araripe within the humid forest enclaves of the Caatinga.

Type locality. Brazil, Ceará, Crato, APA da Chapada do Araripe/FLONA do Araripe-Apodi.

Distribution. Only recorded at the type-locality

*Epiperipatus bouvieri* (*Fuhrmann, 1913*)
(Figs. 12, 13)

*Peripatus bouvieri Fuhrmann, 1913*: 245; *Fuhrmann, 1914*: 186; *Peck, 1975*: 348; *Oliveira, Read & Mayer, 2012*: 24; *Oliveira, 2023*: 172
*Epiperipatus bouvieri*; *Costa, Giribet & Pinto-da-Rocha, 2021*: 790.

Type material: COLOMBIA, Boca del Monte, at the border between Casanare and Arauca.

Examined material. **ICN-ONY-13**, 1 ♀, COLOMBIA, Cundinamarca, Soacha, Vereda San Francisco, Granja Ecológica El Porvenir (2.500 m asl.), 17.viii.2009, Luna, D. leg. **ICN-ONY-27 and 28**, 1 ♀ and 1 ♂, San Antonio del Tequendama, Reserva Los Tunos, 28.v.2012, Chagas-Junior, A. and Chaparro, E. leg. Type material not examined.

Emended diagnosis. Primary papillae with a round, dome-shaped basal piece, separated by an indistinct boundary. Basal piece has 15–18 scale ranks, while the apical piece has 2–3 posterior scale ranks. Males exhibit conspicuous anal glands, represented by two light pores on the anterior margin of the anal aperture.

Redescription. Measurements. Length: 30–56 mm; width 4.2–6.3 mm; height: 2.5–7.0 mm. Color. The examined specimens were discolored, with poorly preserved dorsal diamond patterns. Previous descriptions report a red body background with clear diamond-shaped patterns. Light primary papillae are present but not well-defined.

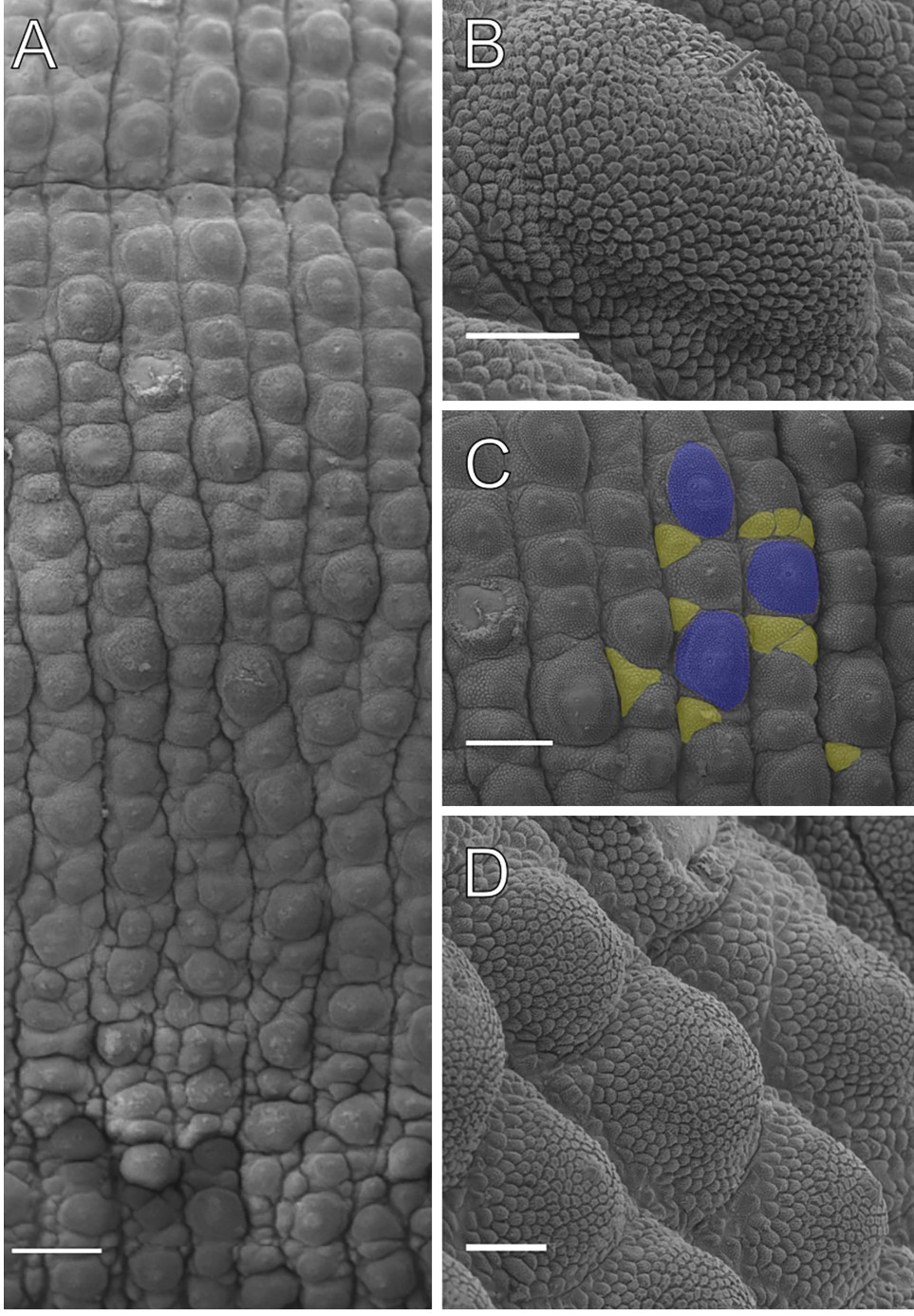

**Figure 12 Dorsal body details of *Epiperipatus bouvieri*.** Scanning electron micrographs of the body of *E. bouvieri*, ICN-ONY=28, Cundinamarca. (A) Dorsal papillae arrangement around the dorsomedian furrow (at the top of the image). (B) Primary papillae in anterior view. Note the large range of scale ranks on the basal and the flat apical piece. (C) Note the primary papillae (highlighted in blue), and the small accessory papillae (highlighted in yellow) present on the flanks of the plicae. (D) Dorsal papillae arrangement. Note the slight difference in size between the primary and accessory papillae. Scale bars in A = 250 µm; B and D = 50 µm; and C = 200 µm.

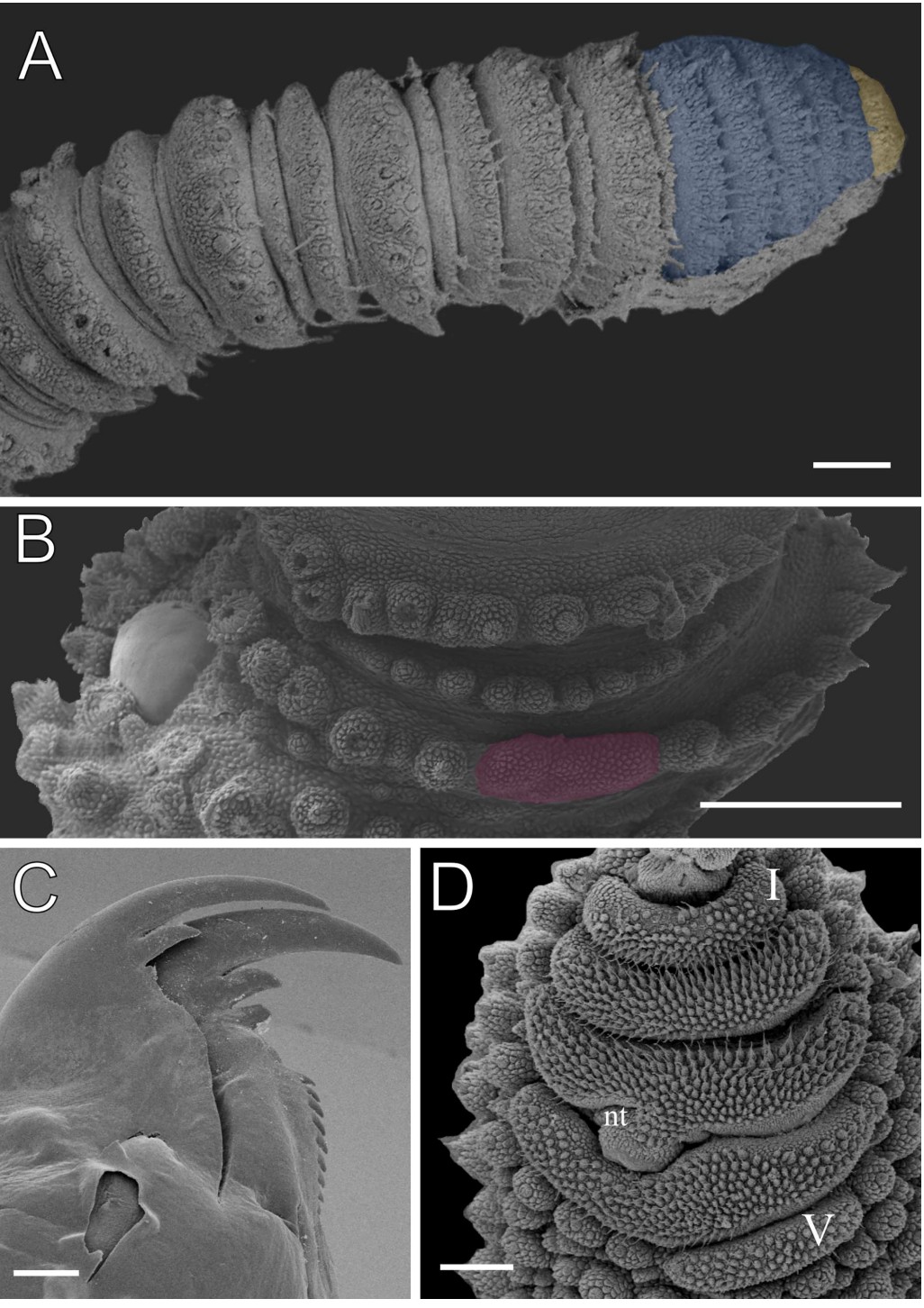

**Figure 13 Head and oncopod parts details of *Epiperipatus bouvieri*.** Scanning electron micrographs of the head of *E. bouvieri*, ICN-ONY = 28, Cundinamarca. (A) Right antenna showing antennal tip (highlighted in yellow) with five large rings (highlighted in blue) followed by the first slim ring and a sequence of broad and narrow antennal rings. (B) Base of the right antenna showing the eye (in the left side) and frontal organ (highlighted in purple). (continued on next page…)

Body description. Due to poor preservation, the dorsal body background could not be fully analyzed. However, a conspicuous dorsomedian furrow and hyaline organs were clearly observed. Each segment features twelve uniformly wide plicae, with seven extending to the ventral side (Fig. 12A). Dorsal papillae have a broad, rounded base, sometimes nearly spanning the entire width of the plicae. Only the smallest accessory papillae are found on the flanks of the plicae and occasionally in the furrows (Figs. 12A and 12C). Primary papillae are always separated by one to three accessory papillae (Fig. 12C). Primary papillae are the largest dorsal papillae and occur exclusively on the tops of the plicae, whereas accessory papillae are present on both the tops and flanks of the plicae (Figs. 12A–12D). Primary papillae possess a rounded, dome-shaped basal piece and a flat apical piece, separated by an indistinct constriction. Basal piece includes 15–18 scale ranks, while the apical piece has 2–3 posterior scale ranks (Figs. 12B and 12D). Posteriorly directed needle-shaped sensory bristle present (Figs. 12B and 12D). Conspicuous ventral and preventral organs.

Head. No distinct structures or patterns are evident on the head. The antennae consist of 45–56 rings. The antennal tip comprises five broad rings (excluding the disc) (Fig. 13A), followed by an alternating sequence of narrow and broad rings up to at least the 20th ring. Eyes and frontal organs are located ventrolateral to the antennal base. The frontal organs are conspicuous and as long as six fused antennal papillae (Fig. 13B). The mouth opening is surrounded by a single small anterior lobe, followed by seven flanked lobes that decrease in size from the anterior to the posterior end of the mouth. The dental formulas for the inner and outer jaws are 1/1-2 and 1/1-2/10, respectively (Fig. 13C).

Legs pairs: 28–33. The nephridial tubercle is present on the fourth and fifth pairs of legs, positioned between the third and fourth spinous pads and connected at the top to the third spinous pad (Fig. 13D). On the fourth and fifth leg pairs, five complete spinous pads are present, with the fifth being the smallest shifted toward the prolateral region (Fig. 13D). Two prolateral and one retrolateral foot papillae are present on the feet of the fourth and fifth pair of legs.

Sexual dimorphism. Males have one pair of pregenital legs, each bearing two crural papillae, being absent in females. Male anal glands are conspicuous, represented by two light pores on the anterior margin of the anal aperture, while absent in females.

Remarks. Legs pairs variation. Males 28; females 28–33. *Epiperipatus bouvieri* is among the largest species in Peripatidae, with specimens reaching up to 56 mm in length. The species is characterized by large basal pieces and flat apical pieces of the primary papillae. Additionally, this is the first documentation of a male, easily identifiable by its prominent anal glands. Previous nomenclatural confusion involving *Peripatus jamaicensis* mut. *bouvieri* (see *Cockerell, 1901*: 326) has been clarified by *Oliveira, Read & Mayer (2012)*. In this study, we adopt the name '*bouvieri*' for the species from Colombia, specifically

in Boca del Monte, as originally described by *Fuhrmann (1913)*. However, *Peck (1975)* reported the occurrence of this species in Cundinamarca, based on a female deposited in the Invertebrate Zoology collection of the Museum of Comparative Zoology (ID: IZ-83639). Onychophorans are generally known for their highly restricted distributions, low dispersal capabilities (*Murienne et al., 2014*), and strong dependence on specific environmental conditions (*New, 1995*). However, recent reports suggest that some taxa may have broader distributions (see 'Discussion'). Comprehensive analyses using phylogenomic data and total evidence approaches to compare the genetic variability of the Boca del Monte and Cundinamarca populations could significantly enhance our understanding of the species. If low molecular variability is observed, it would suggest that both populations belong to the same species, whereas high variability would indicate the presence of more than one species.

Type locality. Colombia, Boca del Monte.

Distribution. Colombia, Aruaca, Casanare (see *Oliveira, 2023*: 172) and Cundinamarca (new records).

## DISCUSSION

Peripatidae and Peripatopsidae have long suffered from a lack of morphological synapomorphies to delimit their internal groups, as noted by *Peck (1975)*, *Read (1988)*, and *Reid (1996)*, which led to the creation of poorly characterized and monotypic genera. In the case of Neopatida, analyses based on extensive molecular and morphological datasets have identified problems with the delimitations of different genera (*Costa, Giribet & Pinto-da-Rocha, 2021*), especially those with presenting great taxonomic diversity and widespread distributions. Currently, experts are making special efforts using combined approaches to distinguish between the two most criticized genera, *Epiperipatus* and *Peripatus* (*Giribet et al., 2018*), due to the unclear morphological boundaries of both, besides the definition of character states such as "crural tubercles in more than two pregenital oncopods, or generally crural tubercles in two pregenital legs", classically proposed the delimitation of the latter (*Bouvier, 1905*: 160, translated herein). As *Epiperipatus* and *Peripatus* encompass the highest species diversity and distribution, they still need to be resolved, considering their considerable variability in commonly used characters, besides the poor preserving methods used (*Read, 1988*). Moreover, *Costa, Giribet & Pinto-da-Rocha (2021)* highlighted the challenges faced in using current morphological characters to support clades within *Epiperipatus*, although some internal clades are still supported by morphology, such as *E. hyperbolicus*, *E. lucerna*, and *E. titanicus* (*Costa, Chagas-Junior & Pinto-da-Rocha, 2018*).

We acknowledge the weakness of classical characters, but there is no need to separate currently described taxa into more groups. Neopatida has been thoroughly discussed considering different biogeographic and phylogenetic approaches in studies on Onychophora (*Giribet et al., 2018*), without any evidence that would change what is currently know about relationships among species. On the other hand, taxonomic changes need to be implemented based on evidence that the recent catalog (*Oliveira, 2023*) has ignored. Furthermore, we recognize that the complexity of node combinations in the

topologies depends on the number of terminals and informative characters used. Thus, including the type species of Peripatidae might or might not change the relationships among terminals, as observed in the last two phylogenetic analyses (*Costa, Giribet & Pinto-da-Rocha, 2021*; *Baker et al., 2021*).

Additionally, there is no plausible phylogenetic justification to support the revalidation of the genera nested within the *Epiperipatus* clade in the recent phylogenies. While acknowledging that the phylogenetic evidence for merging *Cerradopatus*, *Principapillatus*, and the recombination of two species of *Peripatus* into *Epiperipatus* is limited and could be strengthened, it is important to note that the proposed synonymizations align with the evidence currently available. The recent changes, however, may have prioritized nomenclatural stability over consistency with the existing data. In this case, we preferred to keep *E. bouvieri* (*Fuhrmann, 1913*) and *E. sucuriuensis* (*Oliveira et al., 2015*) as proposed in *Costa, Giribet & Pinto-da Rocha (2021)*.

In the case of *E. acacioi*, we had the opportunity to examine the specimens from the collections of Museu Nacional and Universidade de São Paulo, and except for the syntypes, the majority of the analyzed specimens were adequately fixed and well preserved. As in the case of *E. biolleyi*, *E. acacioi* is one of the most studied species of Peripatidae, being revised by *Oliveira, Wieloch & Mayer (2010)* and transferred to *Epiperipatus*. The generic characters of *E. acacioi* do not refute its current taxonomic position, consequently being one of the most representative species of the genus, considering their fit with a combined set of morphological diagnostic features proposed by *Bouvier (1905)* and *Read (1988)*, as discussed by *Costa, Chagas-Junior & Pinto-da-Rocha (2018)*, and supported by the hypotheses presented in *Giribet et al. (2018)* and *Costa, Giribet & Pinto-da-Rocha (2021)*. Additionally, we observed two representative species groups based on the presence of dorsal incomplete and complete plicae, although this did not reflect the recovered phylogenetic relationships. *Epiperipatus acacioi* represents the group of species with incomplete plicae, as also including *E. machadoi* and *E. ohausi*, whereas *E. brasiliensis*, *E. bouvieri*, and *E. cratensis* represent the group with complete plicae, like *E. tucupi Froehlich, 1968*. Thus, given the ongoing debate (*Oliveira, 2023*), while the boundaries between *Epiperipatus* and *Peripatus* remain unresolved, we proceed cautiously, maintaining the results presented here as species currently classified within the genus *Epiperipatus*.

## Implication for the distributions of these species

*Epiperipatus brasiliensis* and *E. cratensis* appear to be restricted to their respective type localities and surrounding regions, a pattern that has been observed in several other Neotropical species and is commonly assumed due to the limited dispersal capability of the onychophorans. This assumption generally holds, except in the cases of distributions expanded by random events, including stepping-stone and over-water dispersals, as observed for *Oroperipatus* sp. from the Galapagos (*Baker et al., 2021*). The poor knowledge of Onychophora is insufficient to reasonably confirm the distribution patterns for all species in the group, as sampling coverage is low, and just a few species have been deeply studied. Therefore, the broad distribution of *E. acacioi* reported here, besides the cases of *Peripatopsis balfouri* (*Daniels, McDonald & Picker, 2013*), *Peripatopsis moseleyi* (*Ruhberg*

*& Daniels, 2013*), and the record of *Oroperipatus* sp. in the Galapagos (which may be a continental species; see *Espinasa et al., 2015*), are among the few currently known cases of species with geographic distributions more extensive than expected for onychophorans.

In addition, enhancing the knowledge of *E. bouvieri* would help to shed light on new occurrence records with a relevant geographical distance from the type locality, as in the case of Boca del Monte in Cordoba and Cundinamarca, approximately 460 km apart from each other. Considering the other cases of broad distribution previously mentioned, *E. bouvieri* may represent one of the rare instances of long-distance dispersal among these invertebrates, which are highly sensitive to environmental changes. Ultimately, this case would also benefit from a focused study based on phylogenomic approaches to clarify the relationship among both populations and adequately assess whether there are two separate species or only one with a wide distribution.

Few Brazilian species of Onychophora are abundant and easily found under decomposing tree logs at the forest edge in disturbed areas, akin to *E. biolleyi* (in Costa Rica), and *Peripatus heloisae Carvalho, 1941*, *E. lucerna*, *E. hyperbolicus*, and *E. titanicus* (in Brazil). Specimens of *E. acacioi* collected at the Estação Ecológica do Tripuí were found in abundance, typically forming clusters under tree logs. Remarkably, they were also observed in association with highly disturbed areas used for solid waste disposal, with females being more numerous than males. This species can be found year-round, although being more commonly sighted towards the end of the rainy season and during the dry season, from April to October (see the redescription of *E. acacioi* above). However, nearly all specimens were collected in areas with high humidity, making these microhabitats veritable refuges for the species during the driest periods. Similarly, specimens collected in the Serra da Moeda were found at the entrances of ferruginous caves, which exhibit higher humidity levels than surface environments.

Concerning its distribution, *E. acacioi* was considered, for decades, endemic and restricted to Ouro Preto, Minas Gerais. However, the new records of *E. acacioi* specimens collected in the Parque Estadual do Itacolomi and caves in the Serra da Moeda, in the Quadrilátero Ferrífero region, reveal a significant expansion in the known distribution of this species (Fig. 1). Despite the new sampling points being located more than 40 km away from the type locality, the Serra da Moeda is within the same environmental context as the Estação Ecológica do Tripuí. Both areas are part of the Quadrilátero Ferrífero, which covers an area of approximately 7,200 km$^2$ and is considered one of Brazil's most important mineral provinces. This region is of great biological importance due to the presence of iron ore fields, the occurrence of endemic plant species, and its unique character in the state of Minas Gerais, situated at the ecotone of two important global biodiversity hotspots, the Cerrado and the Atlantic Forest (*Harley, 1995*; *Giulietti, Pirani & Harley, 1997*).

The ancient and geologically complex terrain of the Minas Supergroup encompasses a variety of lithologies, including dolomites, quartzites, and iron ore deposits. One of the main characteristics of the iron formation is the presence of an extensive underground network, known as canaliculi, which form due to the natural porosity of the rock, especially in the *canga*, a ferruginous sedimentary rock that covers the region's hills (*Alkmim & Marshak, 1998*; *Klein & Ladeira, 2000*). This natural porosity interconnects micro-, meso-, and

macrocavities, creating a widespread network of underground ecosystems (*Ferreira, 2005*; *Zeppelini et al., 2022*). As a result, several endemic species of underground environments, known as troglobites, have a wide distribution in the Quadrilátero Ferrífero, further indicating the existence of these connections that facilitated the dispersal of these species over thousands of years of evolution (*Brescovit et al., 2012*; *Brescovit et al., 2021*; *Brescovit & Sánchez-Ruiz, 2016*; *Cipola et al., 2020*; *Zeppelini et al., 2022*). Therefore, it is likely that this extensively connected network also facilitated the dispersal of *E. acacioi* across different hills in the Quadrilátero Ferrífero region, including the Parque Estadual do Itacolomi and the Serra da Moeda.

## Conservation remarks for the Brazilian Onychophora

Although far from being as "charismatic" as mammals, frogs, and birds, the onychophorans have gained political focus for their role in conservation measures for invertebrates in Brazil. The yield of these actions has benefited, in particular, the critically endangered species from Minas Gerais, such as *E. adenocryptus Oliveira et al., 2011*, and *E. paurognostus Oliveira et al., 2011*. Recognizing the fragility of the populations of this invertebrate group represents an essential step in political actions aimed at conserving the ecosystems associated with these animals. Considering their restricted distribution, limited mobility, and difficulty to adapt to sudden changes in their habitats, the recognition of onychophorans as flagship species is of utmost importance for the development of conservation policies aimed at the usually overlooked invertebrates (*Monge-Nájera, 1995*).

Moreover, the recent update of the official regional red list in the Biodiversity Extinction Risk Assessment System (Sistema de Avaliação do Risco de Extinção da Biodiversidade–SALVE; *ICMBio, 2023*) has added 11 endangered species within protected areas, with *E. acacioi*, *E. brasiliensis* and *E. cratensis* listed among them. Although *E. cratensis* occurs within the sprawling FLONA do Araripe-Apodi, in Ceará, the scarce distribution data prevailed in the decision to include this species. Additionally, this update has revised the count of threatened Brazilian onychophoran species to 19. In this total, these species are currently categorized as follows: six with Data Deficient (DD) status, eight as Least Concern (LC), one as Near Threatened (NT), one as Vulnerable (VU), one as Endangered (EN), and two as Critically Endangered (CR). Thus, the number of species assessed as DD reinforces the urgency of robust studies addressing the still poorly known Brazilian Onychophora.

Regarding the four species examined here, it is crucial to consider their particularities for planning effective conservation measures. First, *E. brasiliensis* was recorded from a humid habitat in the FLONA do Tapajós, in Santarém, Pará. The region is characterized by dense forests with a substantial layer of leaf litter covering the ground, reflecting the total dependence of the species on this type of habitat, similar to other onychophorans. In addition, the soil in the region is marked by a shallow layer of tangled small roots, where the onychophoran was collected 118 years after its first record from 1896 (see *Bouvier, 1905*: 274). This forest is legally protected and covers an extensive area of approximately 530,000 ha, and the current record of a single specimen suggests the possibility of a small population, although this assumption has not yet been tested. Fortunately, there are

no known imminent human threats to indicate a risk of extinction for this population. Therefore, *E. brasiliensis* has been categorized as LC in SALVE based on the IUCN criteria.

On the other hand, *E. cratensis* is apparently endemic and highly dependent on humid and shaded habitats of the "Brejos de Altitude" of its type locality, the APA da Chapada do Araripe, in Crato, Ceará (*Brito et al., 2010*). Frequent anthropogenic pressures in the region include the expansion of livestock ranching, slash-and-burn agriculture, the negative impacts of nature-based tourism, and urban sprawl. Nevertheless, the most common reports are wildfires and cattle presence in the FLONA do Araripe-Apodi, a legally protected area covering an extension of almost 39,000 ha. Despite the pressures from the agribusiness and urban development in the vicinity of its type locality, new specimens have been reported within the FLONA do Araripe-Apodi, thus reducing the risk of endangerment in the near future for this species. Therefore, *E. cratensis* was categorized as LC in SALVE.

Among the Brazilian Onychophora, the case of *E. acacioi* emerges prominently, with the species being previously assessed as EN until 2008, due to its area of occupancy being smaller than 500 km$^2$ within a severely fragmented and continuously decreasing area of occupancy (EN–B2ab(ii)) (https://salve.icmbio.gov.br#/). Currently, its threat level has been reduced to NT (*Costa & Cordeiro, 2023*) thanks to conservation efforts and the accumulation of knowledge on the species, which was the first to receive widespread attention due to its broad applicability in various scientific fields. Consequently, *E. acacioi* became the flagship species for a protected area in Brazil, being distinguished as the only invertebrate to receive such recognition in the country.

Since 1975, monitoring efforts, including studies conducted by *Lavallard et al. (1975)*, have produced significant and abundant data to guide the conservation of *E. acacioi*, permitting insights into its behavioral biology and extended distribution across three locations in Minas Gerais: Estação Ecológica do Tripuí, Ouro Preto (the type locality), which was specifically established in 1978 for the protection of the species; Parque Estadual do Itacolomi, in Mariana and Ouro Preto; and, most recently, in caves of the Serra da Moeda, in Rio Acima, in the Quadrilátero Ferrífero region. The culmination of these efforts, spanning 50 years, revealed the tenacity of the species against threats, providing invaluable insights gleaned from committed conservation programs. Moreover, discovering a new occurrence locality for *E. acacioi* reinforced the notion of an initially underestimated distribution, thus raising the same question for other species. Consequently, an urgent call to action resonates as a necessity to prioritize the conservation agenda for other Brazilian Onychophora. Brazil's official regional red list includes nine species occurring within protected areas (notably, *E. sucuriuensis* is not listed), of which *E. acacioi* is the only one that enjoys a secure status. Thus, recent efforts for a better understanding of the species, including the present work, emphasize the indispensable role of academic and political dedication in safeguarding and preserving not only the velvet worms but all the frequently neglected Brazilian invertebrate biodiversity.

## ACKNOWLEDGEMENTS

We acknowledge the support of Douglas Zeppelini, Estevam Cipriano Araújo de Lima and Bruna Carolline Honório Lopes for the use of the electron microscope laboratory

at the Sistemática de Collembola e Conservação and Instituto de Biologia de Solo at the Universidade Estadual da Paraíba (UEPB), the electron microscope laboratory at Instituto de Biociências of the Universidade São Paulo, the electron microscope laboratory at Museu de Zoologia da Universidade São Paulo, and the Center for Nanoscale Systems, Harvard University for the SEM images.

### Funding

This work was funded by Vale S.A. (Projeto Diversidade Biológica de Cavernas, R100603.CD.0X; Projeto Centro de Triagem de Invertebrados, R100603.CT.0X), Fundação de Amparo à Pesquisa do Estado de São Paulo (FAPESP 2011/20211-0, 2012/02969-6 and 2014/20557-2) to C.S.C. and R.P.R, and Organização de Apoio à Pesquisa da Biodiversidade (OAPBio). The funders had no role in study design, data collection and analysis, decision to publish, or preparation of the manuscript.

### Grant Disclosures

The following grant information was disclosed by the authors:
Vale S.A. Projeto Diversidade Biológica de Cavernas: R100603.CD.0X.
Vale S.A. Projeto Centro de Triagem de Invertebrados: R100603.CT.0X.
Fundação de Amparo à Pesquisa do Estado de São Paulo: FAPESP 2011/20211-0, 2012/02969-6 and 2014/20557-2.
Organização de Apoio à Pesquisa da Biodiversidade (OAPBio).

### Competing Interests

The authors declare there are no competing interests.

### Author Contributions

- Cristiano Sampaio Costa conceived and designed the experiments, performed the experiments, analyzed the data, prepared figures and/or tables, authored or reviewed drafts of the article, and approved the final draft.
- Robson de Almeida Zampaulo conceived and designed the experiments, prepared figures and/or tables, authored or reviewed drafts of the article, and approved the final draft.
- Santelmo Vasconcelos conceived and designed the experiments, performed the experiments, analyzed the data, authored or reviewed drafts of the article, and approved the final draft.
- Michele Molina performed the experiments, analyzed the data, authored or reviewed drafts of the article, and approved the final draft.
- Igor Cizauskas analyzed the data, prepared figures and/or tables, authored or reviewed drafts of the article, and approved the final draft.
- Ricardo Pinto-da-Rocha conceived and designed the experiments, analyzed the data, authored or reviewed drafts of the article, and approved the final draft.

### Field Study Permissions

The following information was supplied relating to field study approvals (i.e., approving body and any reference numbers):

All collected specimens were in accordance with the sampling permits 32882-1 and 34803-2 by ICMBio/MMA (Brazilian Ministry of Environment).

### DNA Deposition

The following information was supplied regarding the deposition of DNA sequences:

The newly sequenced data for the specimens of *Epiperipatus acacioi* from Rio Acima are available at GenBank: PP051244–PP051249 (16S rRNA), PP054357–PP054362 (COI), PP060402–PP060407 (12S rRNA), and PP060417–PP060422 (18S rRNA).

### Data Availability

The raw sequencing data is available at NCBI: PRJNA1074492 and SAMN39861462–SAMN39861467.

### Supplemental Information

Supplemental information for this article can be found online at http://dx.doi.org/10.7717/peerj.19168#supplemental-information.

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
