# Peer review of "Redescription of four Epiperipatus species with an update on the distribution of Epiperipatus acacioi (Marcus & Marcus, 1955)"

_PeerJ, doi:10.7717/peerj.19168_

## Round 0.1 · original submission · Minor Revisions

The three reviewers suggested many minor revisions, while considering the manuscript as valuable. Please consider all of the suggestions in your revised manuscropt.

Reviewer 1 ·

Basic reporting

Good background literature and all the relevant sources cited. Language good and reasonable. Figure professional and of high quality.

Experimental design

Fine

Validity of the findings

Solid interpretation of results.

Additional comments

The phylogenetic tree should have species names in italics.

·

Basic reporting

See below

Experimental design

See below

Validity of the findings

See below

Additional comments

This study provides, among other things, much needed redescriptions of four South American species of Epiperipatus while providing an updated phylogenetic analysis for the group and discussing the conservation status of velvet worms in Brazil. The paper is certainly more interesting than the title suggests, so I would encourage the authors to work on it a bit more, since they have a discussion on nomenclature, phylogenetic analyses, short-read genomic sequence data, conservation, etc. In general, the paper is well written, with some minor language glitches. These are listed below.

I have a few small requests that may improve the paper:

I loved the colored SEM separating primary and secondary papillae, but this is unfortunately only done for one of the four species (a fourth species has some colors for secondary papillae). Why not doing this for all?
When listing species authors after species names, do not use “et al.”, list all authors. I know this is a bit cumbersome, but it is the proper way, I believe (Brito et al., Oliveira et al., etc.)
Lines 103–104: Here you say you have provided codes in parentheses for the colors, but there are no such codes in the descriptions.
It is great that genomic libraries were sequenced, but there is no information about where are they deposited. Please, upload them to SRA and provide the accession codes. Same go for the assembled sequences fished from the libraries.
For E. acacioi, if you examined the syntypes and redescribed the species, why not designating a lectotype and the paralectotypes?
Lines 240–241: You say that the accessory papillae are always separated by two primary papillae”, but this is not what I see in the SEM image. Maybe coloring them, as suggested above, will clarify this?
In the section on Head description for the species, the authors always say “No evidence of structures...”, but heads have “structures” like eyes, antennae, slime papillae... Is there another way to say that they don’t have any other organs? Do you refer to sexually dimorphic organs?
Lines 554–558: I think I know what you mean, but it is a bit unclear. I would also avoid using words like “ignoring” and “dishonesty”; You could say it more simply, that the recent catalogue has ignored taxonomic actions based on evidence and leave it there.

Minor edits
Title: “with an update...” instead of “with update...”
Line 33: “dispersal” instead of “dispersion”
Lines 46–47: “the current state of onychophoran systematics still needs improvement...” instead of “the current state of onychophorans systematics still need improvement...”
Line 54: Sato et al., 2018 is about Kumbadjena, not about Peripatidae; Maybe you mean Giribet et al., 2018?
Line 58: What do you mean “in the museum’s zoology collections”? Which museum? Or do you mean museums in general? If so, it should be “in museum zoology collections”
Line 75: Since you know that there are epigean species, albeit undescribed, I would not say “all known species are epigean”; How about “all described species...” or “all named species...”?
Lines 83–85: I do not understand what do you mean with this sentence: “In the vicinity of...”
Line 102: “flashed circular cameras”? Do you mean a “ring flash”?
Line 105: “conceded” means something different; you should say “provided” or “approved”
Lines 191–192: This was a bit confusing; does this mean that the 64 terminals had no missing fragments? Please, specify.
Line 220: “embryos”, not “embrions”
Line 283: “as found in another species” means only in one species, I guess you mean “as found in other species”, referring to multiple other species?
Line 283: I would suggest to use a more telegraphic mode throughout the descriptions. For example, instead of “The gonopore is located close to the penultimate pair of legs...” I would say “Gonopore located close to penultimate leg pair...”
Line 284: “as all species of” should be “as in all species of”
Line 286: “the male syntype”; there are two male syntypes, so you should say “one male syntype” instead of “the”; and if possible, identify which one. This is where designing a lectotype could be useful.
Line 312: “oncopods pairs” should be “oncopod pairs”
Line 317: The furrow starts “in front” of the head? Or “posterior” to the head?
Line 328–329: I can count four to five scale ranks in the SEM image provided.
Line 345: “Four spinous complete pads” should be “four complete spinous pads”
Line 396: “Almost dorsal papillae on plicae”? Do you mean that “almost all dorsal papillae on plicae”?
Line 401: “Basal and the apical piece show...”; Why do you use the definite article for apical but not for basal? Anyway, in telegraphic style it should say “basal and apical piece show...”
Line 424: I don’t see the foot papillae in the SEM, so don’t cite Fig. 11C. Or if I am missing it, then it may be good to indicated them in the figure.
Line 446: What do you mean “with no indication of the holotype”? That the specimen is not labelled as a holotype? How about “with no indication that the specimen is a holotype”
Line 464: There is an underscore in a comma.
Line 495: Do you have any images of the conspicuous frontal organs? It would be really nice to see these.
Line 511: “It is worth noting” is the formal way to write it. “It’s” is colloquial.
Line 550: Again, Sato et al., 2018 is not the correct citation here, as it refers to Peripatopsidae; it should be Giribet et al., 2018.
Line 567: 2023 should be in parenthesis.
Lines 571–572: Make it simpler, just “were well preserved” and “in addition of being properly fixed”.
Line 573: Missing period after the abbreviation of Epiperipatus.
Line 600: Espinasa, not Spinasa.
Lines 612 and 616: I would say “under” or “underneath” instead of “beneath”
Line 632: “Iron fields” should be “iron ore fields” (iron ore is the mineral, iron is the metal once it has been processed). Please, make sure you change this throughout, as you use “iron” several times in the discussion.
Line 637: “iron formation rocks” could be “iron ore deposits”?
Line 685: “anthropic” is not a word, it should be “anthropogenic”
Line 696: I checked this and could not find the species in the list.
References: Please, check italics, which are missing in the titles; Also, some titles are with all words in capitals while others aren’t. Follow the format of the journal throughout.

Looking forward to seeing this paper published!

·

Basic reporting

The English language is generally clear in meaning, but there are many small grammatical, typographical and stylistic errors (see additional comments). The introduction describes the background well to show the context. The literature is well-referenced and relevant. The structure conforms to PeerJ standards and the discipline norm. The figures are relevant and of high quality, but I have made simple suggestions for further enhancement (see additional comments). Raw data is supplied via GenBank (NCBI). It is self-contained with results relevant to hypotheses.

Experimental design

The manuscript presents original primary research within the scope of the journal. The research question is well defined, relevant, and meaningful. It is stated how the research fills an identified knowledge gap. Rigorous investigation is performed to a high technical and ethical standard. The methods are described with sufficient detail and information to replicate.

Validity of the findings

All underlying data have been provided; they are robust and statistically sound. Conclusions are linked to original research questions, but editing is required to ensure the manner of expression is appropriate (see line 589 in additional comments).

Additional comments

Specific recommendations listed below, by line number:

41: The correct citation for Peripatidae is not ‘Evans, 1901’ as long thought, but instead ‘Audouin & Milne-Edwards, 1832’. See de Sena Oliveira (2023).

46, 57: It is the known diversity that has increased.

47: Change ‘eighteen’ to ‘18’, in accordance with the journal’s style.

49: Singularise ‘onychophorans’ and change ‘need’ to ‘needs’.

54: It is the most diverse genus within Peripatidae.

54, 599: In fact, you could replace ‘Peripatidae’ with ‘Onychophora’ and this sentence would still be true.

62: Change ‘museum’s’ to ‘museums’’ (i.e., move the apostrophe).

96: Add ‘(SEM)’ to first mention of scanning electron microscopy.

116: Change ‘following mostly’ to ‘mostly following’.

117: I believe the semicolon should be changed to a comma.

129: Change ‘morphologic’ to ‘morphological’ for consistency.

125: Change ‘scanning electron microscopy (SEM)’ to ‘SEM’.

144: Change ‘1-10’ to ‘1–10’ and ‘extracted DNAs’ to ‘DNA extracts’.

147: Change ‘2×’ to ‘2 ×’.

165: Sampling one tree out of every 1,000.

168: I believe it is conventional to write ‘standard deviation’ rather than ‘sigma’.

169: Change ‘98Ma’ to ‘98 Ma’ for consistency.

186: Capitalise ‘neotropical’.

192: To help readers less familiar with the Onychophora, I recommend indicating the central African distribution of M. tholloni here.

240, 243–245, 247, 400: Unless juvenile or poorly preserved, it would be ideal to determine the sexes of the unsexed specimens.

254: ‘pairs of legs’ is redundant and can be deleted.

257: ‘a little bit’ is informal. I recommend ‘slightly’ instead.

259: Change ‘band light yellowish pink extended by’ to ‘light yellowish pink band extended along’.

260: Change ‘) (’ to a semicolon.

270, 329, 617: If there is no difference between ‘plicae’ and ‘folds’, I recommend using consistent terminology to prevent confusion.

272, 275–276: Instead of ‘both’, ‘both kinds of’ or ‘all’ would be clearer.

275: Change ‘them faded which are the largest one’ to ‘the largest ones faded’.

277: Change ‘In’ to ‘At’.

293–294: Change ‘could be present rarely on the specimens’ to ‘may rarely be present’.

296: Change ‘and’ to ‘with’.

297, 574–575: Change ‘in’ to ‘on’.

303: Instead of ‘present with crural papilla and one crural papilla on each’ would it be simpler to write ‘each with one crural papilla’?

304: By ‘board’ do you mean ‘border’?

305: Singularise ‘pairs’.

309: Change ‘municipalitiesof’ to ‘municipalities of’.

336: Change ‘small diamonds strong reddish brown’ to ‘small, strong reddish brown diamonds’.

348, 426: Pluralise ‘range’.

356: Delete the first ‘the’.

358, 526: Change ‘anterior unique’ to ‘and unique anterior’.

362: Delete the period following ‘species’.

364–365: Change ‘spinous complete’ to ‘complete spinous’.

382–383: Change ‘that E. brasiliensis belong to a new species’ to ‘E. brasiliensis belong to a new species and’.

419: I do not understand the sentence that starts here.

421: Change ‘at’ to ‘across’.

423: Change ‘asymmetrical spherical’ to ‘asymmetrical, spherical,’.

431: In arrangements…

439: Change ‘the 25 ring’ to ‘ring 25’.

445: Elsewhere you have used ‘and’ rather than ‘–’ between immediately consecutive figures. This should be consistent one way or the other.

448: Dehyphenate ‘retro-lateral’ for consistency.

452–453: The sentence that starts here can be simplified to ‘Males are equal to or smaller than females’.

453: Add a decimal place to ‘44’ for consistency.

455, 535: Delete ‘(male)’ or move it to end of clause.

464: It became a locality supposedly shared by two species.

471: Capitalise ‘upon’.

505: Singularise ‘diamonds’.

513: Change ‘on’ to ‘in’.

538: Pluralise ‘male’ for consistency.

539: Change ‘, which’ to a semicolon.

542: Expand ‘It’s’ to ‘It is’.

565: …to delimit their internal groups.

570, 642: Pluralise ‘distribution’.

575: Change ‘my’ to ‘our’, write ‘translated herein’, or indicate which author is responsible for the translation, as there are multiple authors.

587: Pluralise ‘relationship’.

588: The sentence that starts here needs to be qualified. Does ‘no evidence’ refer only to this study or to all studies? Does ‘relationships among species’ refer to species within Onychophora, Peripatidae, Neopatida, or Epiperipatus?

589: (I adopt a position of neutrality on this issue.) I think it is proper for you to disagree without commenting on personal characteristics. Instead, I suggest acknowledging the assertion that the phylogenetic case for sinking Cerradopatus and Plicatoperipatus into Epiperipatus is limited and needs to be strengthened, but follow this by explaining that the synonymisation is nonetheless consistent with the evidence currently available, and that, in your opinion, the recent changes were overly conservative in placing greater value upon nomenclatural stability than consistency with the available evidence.

595: To help readers less familiar with the Onychophora, I suggest being explicit about which names you are regarding as synonyms.

601–602: Why randomly mention distribution in the middle of this section about taxonomy?
It could be moved to the introduction.

605, 606: ‘pretty’ is informal. I recommend ‘quite’ or ‘fairly’ instead.

607: Change ‘such as E’ to ‘as is E.’.

626: Change ‘regions, which have’ to ‘regions; a pattern that has’.

627: Change ‘are’ to ‘is’ and ‘assumed due’ to ‘assumed to be due’.

629: What random events? Could you be more specific?

651: Change ‘doTripuí’ to ‘do Tripuí’.

679: Change ‘micro, meso’ to ‘micro-, meso-’.

690: Change ‘a “charismatic group”’ to ‘as “charismatic”’.

697: Onychophorans are less able to flee unfavourable conditions, but I doubt they are unable to adapt to habitat change. Is this supported by any study? Surely all organisms are able to adapt to changing conditions, at least to some degree.

725: Change ‘shade’ to ‘shaded’.

757: Change ‘from’ to ‘of’.

765: Decapitalise ‘Laboratory’.

Many SEM Figs: I think several micrographs can be enhanced by increasing contrast. Change ‘Scales bars’ to ‘Scale bars’. It is much more common to write ‘Roman numerals’. When defining scale bar lengths, your main figures use semicolons where your SI figures use commas – this should be consistent.

Fig. 1: I do not understand the three rectangles in the key. The two solid lines are too difficult to distinguish from each other, and there are no dashed lines anywhere on the maps. Also, they are listed under the heading ‘Epiperipatus’. I think they can be deleted without any significant loss to interpretation of the key facts. Hyphenate ‘bottom right’.

Fig. 2: Increase the size of the ‘L’ line to be consistent with the other lines. Change ‘updates of’ to ‘updates to’.

Fig. 4: Change ‘near to the’ to ‘near the’ and hyphenate ‘upper right’.

Fig. 5: Hyphenate ‘lower right’.

Fig. 6: Change ‘showing’ to ‘shows’.

Fig. 7: Change ‘dorsomediam’ to ‘dorsomedian’, and ‘on anterior’ to ‘in anterior’.

Fig. 8: Increase the brightness of part B to be consistent with part A. Delete ‘microscopy’.

Fig. 10: Change ‘dorsomedial’ to ‘dorsomedian’ for consistency.

Fig. 12: Decapitalise ‘Papillae Arrangement around the Dorsomedian Furrow’.

Fig. 13: Hyphenate ‘upper right’. Change ‘Scales in bars’ to ‘Scale bars’.

SI Fig. 1: ‘Epiperipatus’ can be abbreviated three times in the caption.

SI Fig. 2: Change ‘segment on’ to ‘segment in’ and decapitalise ‘Primary’.

SI Fig. 3: Change ‘dash’ to ‘dashed’ and ‘of the right’ to ‘on the right’.

---

## Round 0.2 · accepted · Accept

All the suggestions from the reviewers have been taken into account and the manuscript has been greatly improved.